# Autotoxin-mediated latecomer killing in yeast communities

**Arisa H. Oda**[1]\*, **Miki Tamura**[1], **Kunihiko Kaneko**[1,2,3], **Kunihiro Ohta**[1,2], **Tetsuhiro S. Hatakeyama**[1]\*

**1** Department of Basic Science, University of Tokyo, Tokyo, Japan, **2** Research Center for Complex Systems Biology, Universal Biology Institute, University of Tokyo, Tokyo, Japan, **3** The Niels Bohr Institute, University of Copenhagen, Copenhagen, Denmark

\* odar@bio.c.u-tokyo.ac.jp (AHO); hatakeyama@complex.c.u-tokyo.ac.jp (TSH)

**Data Availability Statement:** All sequence data are available from the DDBJ with DRA BioProject (accession numbers PRJDB11860, PRJDB10422; https://ddbj.nig.ac.jp/resource/bioproject/PRJDB10422 https://ddbj.nig.ac.jp/resource/

## Abstract

Cellular adaptation to stressful environments such as starvation is essential to the survival of microbial communities, but the uniform response of the cell community may lead to entire cell death or severe damage to their fitness. Here, we demonstrate an elaborate response of the yeast community against glucose depletion, in which the first adapted cells kill the latecomer cells. During glucose depletion, yeast cells release autotoxins, such as leucic acid and L-2keto-3methylvalerate, which can even kill the clonal cells of the ones producing them. Although these autotoxins were likely to induce mass suicide, some cells differentiated to adapt to the autotoxins without genetic changes. If nondifferentiated latecomers tried to invade the habitat, autotoxins damaged or killed the latecomers, but the differentiated cells could selectively survive. Phylogenetically distant fission and budding yeast shared this behavior using the same autotoxins, suggesting that latecomer killing may be the universal system of intercellular communication, which may be relevant to the evolutional transition from unicellular to multicellular organisms.

## Introduction

When organisms face crises, such as starvation, they are forced to adapt at both individual [1,2] and population levels [3–5]. In unicellular organisms, the former has been intensively studied as an adaptation phenomenon [6–8], whereas the latter is poorly understood. Severe conditions decrease the carrying capacity [9], and unicellular organisms have to decrease the population size [10,11]. However, such an adjustment in cell number carries the risk of killing clonal cells. Thus, how cellular communities adapt to crises without decreasing the fitness of clonal cells remains unknown.

Here, we report an elaborate survival behavior in crisis. When fission yeast, *Schizosaccharomyces pombe*, is cultured under glucose-limited conditions, where the carrying capacity is expected to decrease, cells release toxic molecules into the medium. Such a medium kills even the clonal cells of the toxin-producing cells when they are transferred from glucose-rich conditions. This may look like mass suicide at first glance. However, cells precultured in glucose-depleted conditions continue to grow even in the conditioned medium, as they adapt to the

bioproject/PRJDB11860 All other relevant data are within the paper and its Supporting Information files.

**Funding:** This work was partially supported by the Ohsumi Frontier Science Foundation (https://www.ofsf.or.jp/), the Basic Science Research Projects of Sumitomo Foundation (http://www.sumitomo.or.jp/e/), Japan Society for the Promotion of Science (JSPS) (https://www.jsps.go.jp/english/) KAKENHI (19K16070) to A.H.O., by JSPS KAKENHI (17H06386) to K.K., by JSPS KAKENHI (20H04862) to T.S.H., by Japan Science and Technology Agency (JST) (https://www.jst.go.jp/EN/) CREST, Japan, Grant Number JPMJCR18S3, Japan Agency for Medical Research and Development (AMED) (https://www.amed.go.jp/en/) Grant Number JP20wm0325003 to K.O. The funders had no role in study design, data collection and analysis, decision to publish, or preparation of the manuscript.

**Competing interests:** I have read the journal's policy and the authors of this manuscript have the following competing interests: Patent applications have been filed for the technology described in this publication. A.H.O., T.S.H., and K.O. are named as the inventors of these patents. The remaining authors declare no competing interests.

**Abbreviations:** AnnV, annexin-V; CE-MS, capillary electrophoresis mass spectrometry; CM, conditioned media; gDNA-seq, genomic DNA sequencing; InDel, insertion–deletion; LC-MS/MS, liquid chromatography-mass spectrometry; MM, minimal media; OD, optical density; PI, propidium iodide; WT, wild-type.

toxins through glucose depletion, and such cellular state is inherited. In other words, cells autonomously differentiate into 2 types, adapted and nonadapted ones, and the cellular community selectively saves the former. Yeast cells in glucose-depleted media release toxins, which prevent an invasion of latecomers by killing them, as the Greek philosopher argued: the plank of Carneades [12]. Surprisingly, the same behavior was observed in the budding yeast, which are phylogenetically distant relatives of the fission yeast [13]. We also demonstrate results implicating the underlying molecular mechanisms of the latecomer killing.

## Results

To detect interactions in the population, we prepared conditioned media (CM) by culturing wild-type (WT) fission yeast *S. pombe* referred to as WT CM. First, cells were cultured for 30 h in the minimal media (MM) without glucose, with 3% glycerol as a carbon source (see S1 Fig), and with 0.05% ethanol to induce genes to metabolize glycerol [14] (0% MM). Of note, we added 3% glycerol to every medium we used unless otherwise noted. Next, cells, which had been precultured in the MM with 3% glucose (3% MM), were cultured in the WT CM (see also Fig 1A for the procedure). Then, they stopped growing for approximately 20 h and resumed growing again (see the red line in Fig 1B). We termed this prolonged lag phase as the delay phase (see Text A in S1 Text for measurement of the delay phase). If incubation time to prepare CM was longer than 15 h, such media also induced the delay phase, while shorter incubation time did not introduce such a phase (Fig 1B). Although here we added leucine, adenine, uracil, and histidine to the MM as supplements for auxotroph mutant strains, the delay phase was induced regardless of whether we added these compounds or not (see S2 Fig). It indicated that the delay phase was not caused by the amino acid–rich environment. Those results indicated that in the early growth phase, cells might release inhibitors for growth or depleted some of the nutrients required for such a phase.

To determine whether cells release inhibitors or deplete essential nutrients, we constructed a conditioned medium using a 1,6 bis-phosphatase deletion mutant (*fbp1Δ*), which did not have a functional gluconeogenetic pathway [15]. Such a mutant strain could not grow without glucose (S3 Fig and [14,16]) and was expected not to consume the nutrients required for growth. The CM made using *fbp1Δ* cells (*fbp1Δ* CM) also caused the delay phase (Fig 1C), as shown with the WT CM. This suggested that the delay phase resulted from the release of inhibitory molecules by cells rather than the depletion of nutrients.

In addition, when we administered glucose at a sufficient concentration to the CM (Fig 1D) to recover the carrying capacity, cellular growth was not disrupted, and the delay phase was not observed, i.e., the growth curve of cells in such media was almost the same as that of those in MM with glucose (Fig 1E). This indicated that inhibitory molecules in the CM worked only in the absence of glucose. When we added other species of sugars, i.e., fructose, galactose, and mannose, they were able to reduce the length of the delay phase, but each of them showed different strength of effect; fructose had the same effect as glucose, and galactose and mannose reduced the delay phase only partially (S4A Fig). Furthermore, if we administrated 2-deoxy-D-glucose, a glucose analog not metabolized by the glycolysis, cells did not grow. These results suggest that the rescue of the growth was due to the influence of each sugar on the metabolism, and not to signaling by glucose. Further, when glutamate or a mix of 12 amino acids commonly used as supplements for yeast was added to the CM, they did not rescue the growth (S4B Fig).

After the delay phase, the growth rate in the CM returned to almost the same level as that in the glucose-depleted MM. This suggested that the cells were able to adapt to inhibitory molecules in the CM, and this adapted state was inherited; if cells do not memorize the adapted

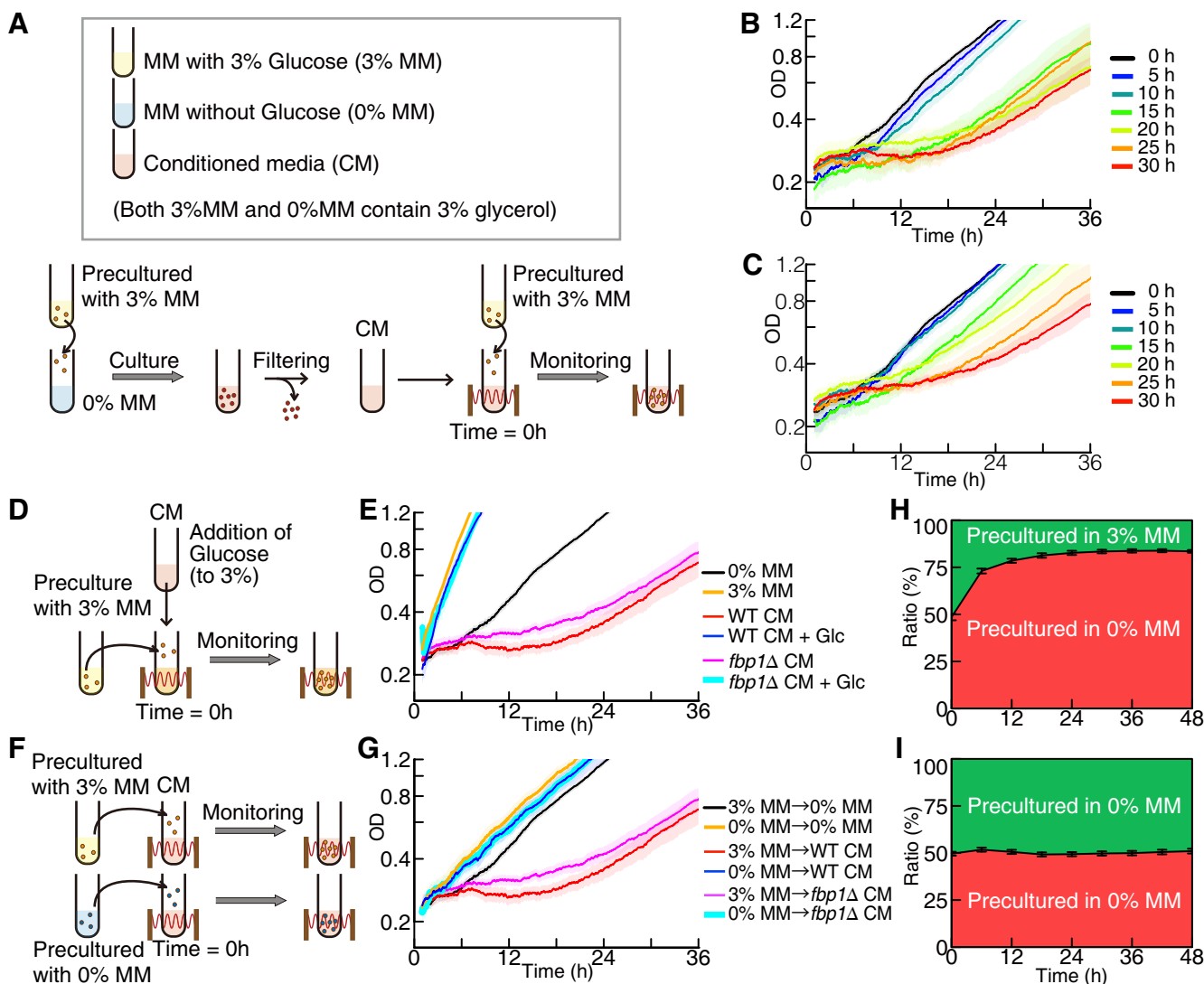

**Fig 1. Conditioned media caused a delay phase of cell growth and latecomer killing during glucose depletion.** (**A**) Schematic illustration of the experimental procedure for (**B**) and (**C**). (**B** and **C**) Growth curves of WT cells in (**B**) WT or (**C**) *fbp1Δ* CM without glucose. Different colored lines indicate a moving average of OD measured every minute in CM with different incubation times. Each line is an average of $n \geq 10$ samples, and the pale-colored area indicates the SEM. (**D**) Schematic illustration of the experimental procedure for (**E**). (**E**) Growth curves of WT cells in CM with 3% glucose. Each line represents an average of $n \geq 7$ samples. (**F**) Schematic illustration of the experimental procedure for (**G**). (**G**) Growth curves of WT cells, precultured without glucose, in the CM. Each line represents an average of $n \geq 7$ samples. (**H** and **I**) Competition assay in WT CM (**H**) between the cells precultured in 3% and 0% MM and (**I**) between the cells precultured in 0% MM. Green and red areas indicate the fraction of mNeonGreen- and mCherry-labelled cells, respectively, and overwriting outline characters indicate preculture conditions. Black vertical bars between 2 areas indicate SEM (the number of each sample is 12). The data underlying this figure can be found in S1 Data. CM, conditioned media; MM, minimal media; OD, optical density; SEM, standard error of the mean; WT, wild-type.

state, divided cells will not survive after the delay phase, and only a small portion of cells can escape death and grow. Then, the growth rate would decrease eventually. However, after the delay phase, the growth rate in the CM was stably maintained. Hence, those growing cells should inherit the adapted state from those mothers.

To verify the existence of the adapted state of cells, we precultured cells in the 0% MM for 24 h and measured their growth in WT and *fbp1Δ* CM (Fig 1F), and a delay phase was not observed (Fig 1G). Furthermore, to verify whether adaptation to the inhibitory compounds

was due to genetic or not, we precultured the cells, which survived in the CM, in 3% MM and, again, cultured them in the CM; they showed a delay phase again (S5 Fig). Moreover, we performed genomic DNA sequencing (gDNA-seq) of surviving cells and identified no unique single-nucleotide polymorphism or insertion–deletion mutations (InDels), except for highly repetitive sequence loci, such as telomeres and centromeres (see S6 Fig and Text B in S1 Text for details). This indicated that the adapted state was inherited without genetic changes.

The plausible evolutionary significance of the release of inhibitory molecules and adapting to them is the inhibition of the growth of different lineages of cells. When sugars around cells are depleted, they start to release inhibitory molecules while simultaneously adapting to such inhibitors. Then, the modified environment will inhibit the growth of latecomers, even if they are closely related. We performed a competition assay by artificially mimicking the above conditions; we simultaneously added cells that were precultured in glucose-rich and glucose-depleted media into the CM at an equal amount in the beginning and observed their population dynamics (S7 Fig). Then, the fraction of adapted cells to unadapted cells continued to increase for 24 h and reached a steady-state (Figs 1H and S8A), while the 50–50 ratio was maintained in the competition assay between adapted cells (Figs 1I and S8B). In addition, the steady-state ratio of adapted and unadapted cells agreed with the ratio predicted from the growth curve observed in fresh and CM shown in Fig 1G (see also Text C in S1 Text). This implied that the combination of inhibitor release and adaptation caused population dynamics shown in Fig 1H and 1I and selected the offspring of inhibitor-producing cells to survive.

The characteristics of the inhibitory molecules observed in the CM helped us isolate them. Since treatments of the CM with autoclaving, DNase, RNase, and proteases did not affect its ability to cause the growth delay (S9 Fig), the inhibitory molecules in the CM should not be peptides, proteins, or nucleic acids, as well as volatile/thermolabile materials. Then, we identified chemical compounds in the freshly prepared MM, as well as WT and *fbp1Δ* CM, using capillary electrophoresis mass spectrometry (CE-MS). We identified 20 chemical compounds (see Table A in S1 Text). From these candidates, we chose 12 chemicals that were included in both types of CM but not the fresh medium (see the yellow hatching region in the Venn diagram in Fig 2A), because both types of CM initiated the delay phase. We further narrowed down 12 molecules by adding them as a single dose to the MM according to the following criteria: (1) They did not change the growth rate significantly after the delay phase. (2) They had little effect on growth in the presence of glucose. (3) They caused a shorter delay phase in cells that had already adapted to glucose depletion. Finally, we isolated 2 small molecules with similar structures: leucic acid (HICA; Fig 2B) and L-2keto-3methylvalerate (2K3MVA; Fig 2C). Note that some of the molecules in the candidate list were difficult to obtain commercially and could not be tested.

The 2 inhibitory molecules had similar characteristics. When the concentration of these molecules was not sufficient, we did not observe the delay phase (Fig 2D and 2E). Then, the more inhibitors we administrated, the longer the delay phase took. Finally, if the concentration was higher than the critical concentrations (30 mM for HICA and 25 mM for 2K3MVA), cell growth was thoroughly repressed. Notably, there are 2 optical isomers of HICA, which are indistinguishable via CE-MS, both of which caused a growth delay at the same concentration (S10 Fig). When glucose was added to the MM simultaneously with inhibitory molecules, cell growth was not disrupted (see the orange line in Fig 2F and 2G). Moreover, even under the administration of such high concentration where cells stopped growing, cells that had been adapted to glucose depletion grew (see blue line in Fig 2F and 2G). These correspondences of inhibitory molecules with the CM implied that the release of HICA and 2K3MVA was one of the causes of growth inhibition during glucose depletion. Since adding leucine or the branched amino acids to the 0% MM did not cause the delay phase (S11 Fig), the growth inhibition was

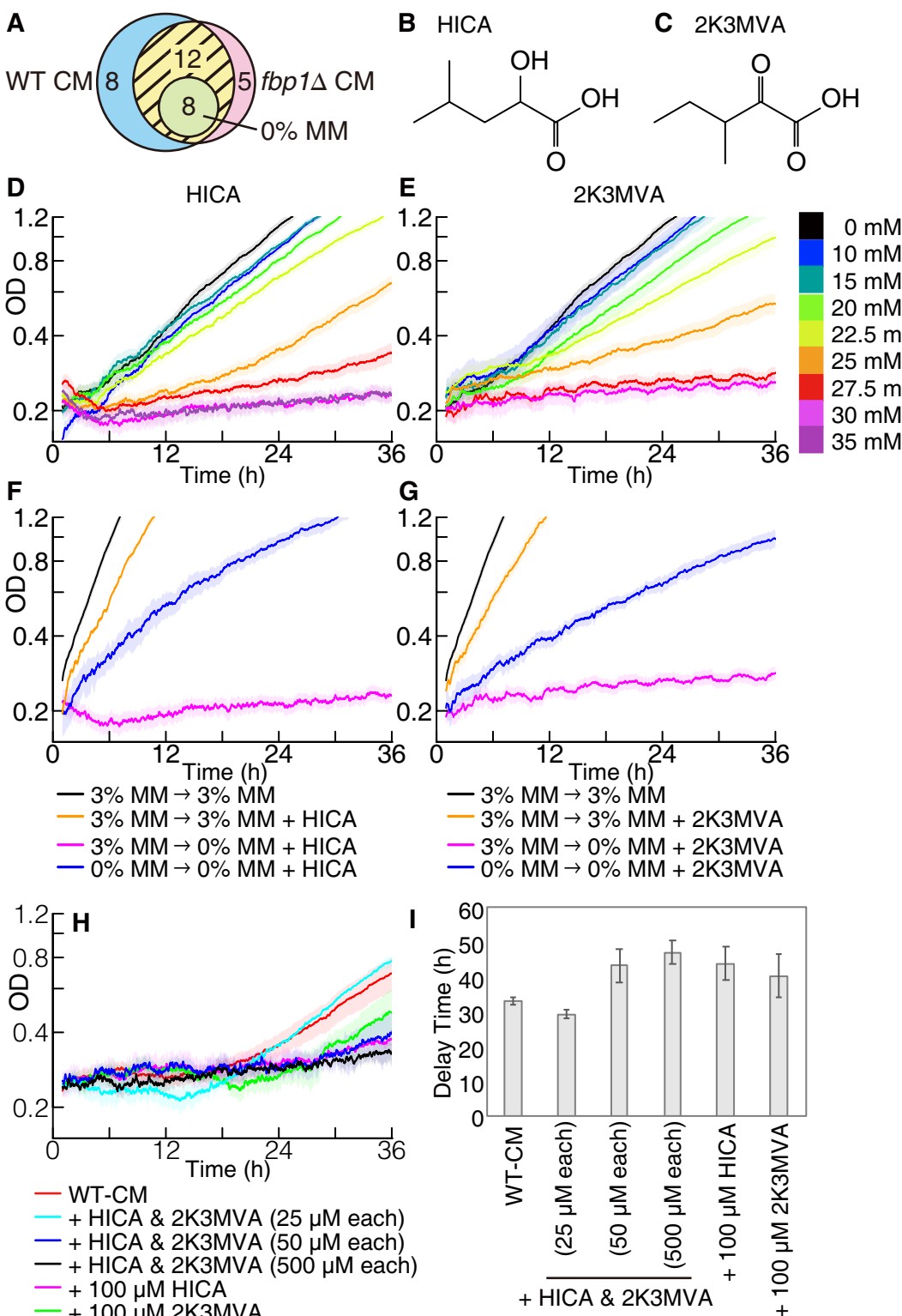

**Fig 2. Identification of growth inhibitors.** (**A**) Venn diagram of compounds detected using CE-MS. Compounds in 0% MM, WT CM, and *fbp1Δ* CM were analysed. Twenty compounds were detected in both WT and *fbp1Δ* CM, and 8 of those were also detected in 0% MM. Thus, 12 compounds (hatching area) were detected uniquely in both types of CM (see Table A in S1 Text for details of the detected molecules). (**B** and **C**) The structure of (**B**) HICA and (**C**) 2K3MVA. (**D** and **E**) Growth curves in 0% MM

in the presence of (**D**) HICA and (**E**) 2K3MVA. WT cells precultured in 3% MM were transferred to 0% MM with various concentrations of the inhibitory compound at 0 h. Each line represents the average of $n \geq 6$ samples. (**F** and **G**) Effects of adaptation and glucose administration on growth curves in the presence of (**F**) 30 mM HICA or (**G**) 25 mM 2K3MVA. The blue line indicates the growth curve of WT cells precultured in 0% MM in 0% MM with the inhibitory compound. The orange line is a growth curve of WT cells in MM with the inhibitory compound and 3% glucose. Pink and black lines indicate growth curves in 0% MM with the inhibitory compound and 3% MM as controls, respectively. Each line represents an average of $n \geq 4$ samples. (**H** and **I**) Cells precultured in 3% MM were inoculated to WT CM with additional HICA, 2K3MVA, or the mixture. (**H**) Growth curves and (**I**) length of the delay phase $\tau$ were plotted. A line for WT CM represents an average of $n = 22$ samples, and those for WT CM with the growth inhibitors represent an average of $n = 3$ samples. The data underlying this figure can be found in S1 Data. CE-MS, capillary electrophoresis mass spectrometry; CM, conditioned media; MM, minimal media; WT, wild-type.

not due to ketogenic amino acids but to specific molecules we identified. Although the quantified concentrations of HICA and 2K3MVA in the CM were rather lower than the effective dose of each compound when only each was added to the medium (see Table B in S1 Text for the concentrations of those in the CM), the addition of HICA and 2K3MVA to the CM drastically reduced the effective dose by approximately 3 orders of magnitude (see Fig 2H and 2I). The addition of HICA and 2K3MVA at the physiological concentration (50 to 100 $\mu$M) to the WT CM led to drastic elongation of the delay phase, suggesting that HICA and 2K3MVA are indeed the important entities of the growth inhibitors that can work in a single dose, while they need other components for their full actions. It suggested that a combination of multiple secreted compounds might synergistically influence, while the main substances of growth inhibitors are HICA and 2K3MVA since other compounds detected did not cause the delay phase solely. The concentration of those molecules increased with time; from 10 to 20 h, they increased more than 5 times, and from 20 to 30 h, they increased more than 3 times. Accordingly, the delay phase appeared in the CM with a longer incubation time than 15 h (Fig 1B). It is consistent with the characteristics of the identified molecules, which repressed the cell growth above the critical concentrations (Fig 2D and 2E). In addition, since the cell doubling time in 0% MM was about 10 h (Table C in S1 Text), the concentrations of those molecules increased faster than cells, indicating that those molecules rapidly accumulated in the media. Note that the concentrations of identified growth inhibitors in *fbp1*Δ CM were lower than those in WT CM, but those 2 CMs showed almost the same length of the delay phase (Table C in S1 Text). It implies that the *fbp1*Δ CM contains substantial amounts of cofactors for growth inhibitors that are not effective alone but can enhance the action of HICA and 2K3MVA. Even if such compounds exist, they will not be observed by the method we adopted here, and future studies will be necessary to identify them.

We next explored the synthetic pathway of HICA and 2K3MVA. Although a synthetase for HICA (hydroxyisocaproic acid dehydrogenase) was identified in bacteria (see [17] for *Lactococcus lactis*), neither its homolog nor any enzymes for HICA and 2K3MVA synthesis have been reported in *S. pombe*. We, therefore, searched for genes that encode hydroxyacid dehydrogenases, whose targets are unknown in *S. pombe*, by using UniProt [18], and made their disruptant mutants. Then, the concentrations of HICA and 2K3MVA in media conditioned by all of these strains decreased but did not reach zero (e.g., 5.3 to 10.6 $\mu$M of HICA from 55.2 to 68.1 $\mu$M in WT measured in the same condition; Table B in S1 Text). Correspondingly, these CM caused a shorter delay phase than that caused by the WT CM (see S12 Fig). These results suggest that *S. pombe* synthesized HICA and 2K3MVA by using multiple pathways and synthetases. More importantly, adding back 25 to 50 $\mu$M of HICA or 2K3MVA, equivalent to their concentrations in the WT CM, to media conditioned by those mutants rescued the delay phase (see S13 Fig). These results again support that HICA and 2K3MVA are real substances of growth inhibitors and unidentified minor compounds in CM promote the action of HICA and 2K3MVA.

How do the inhibitory molecules cause the delay phase? There are 2 possible mechanisms: delay of initiation of growth in each cell or death of the majority of cells. In the latter, the concentration of living cells will be masked by that of dead cells in the optical density (OD) measurement, and an apparent delay phase will be observed until the concentration of living cells exceeds that of dead cells (see Text D in S1 Text and S14 Fig). To verify which hypothesis is correct, we counted the number of dead cells by staining them with phloxine B [19] (Fig 3A–3H). Then, over 80% of cells, which were cultured in the presence of a higher concentration of inhibitory molecules, were dyed in red (Fig 3G and 3H). Indeed, the majority of cells during the delay phase were stained red and showed a typical rod shape, as observed under a microscope, which showed the characteristics of dead cells [20,21] (Fig 3E and 3F). In contrast, only a small number of cells showed a spherical shape similar to cells cultured in MM without glucose. This indicated that only a small portion of the cells survived and continued to divide in the presence of inhibitory molecules. Similarly, the death rate in WT and *fbp1Δ* CM increased (Fig 3C, 3D and 3G). Furthermore, cells precultured under the glucose-depletion condition, which did not show a delay phase in the presence of the inhibitory molecules, were mostly alive (Fig 3H). This suggested that HICA and 2K3MVA induced cell death, which was the primary cause of the delay phase (see also Text D in S1 Text, S15 and S16 Figs and Table C in S1 Text).

We analyzed how cells were killed by the autotoxins or CM by costaining of annexin-V (AnnV) and propidium iodide (PI): Early apoptotic cells exhibit phosphatidylserine externalization and are stained by AnnV but not by PI. In contrast, primary necrotic cells show ruptured plasma membranes, which are stained by PI but not by AnnV [22]. Late apoptotic and secondary necrotic cells are stained by both AnnV and PI. As a result, cells in the autotoxins and CM caused cell death via 2 processes, i.e., apoptosis and necrosis, as shown in S17 Fig. It suggests that the mechanism of cell death is not simple but multifarious, and there is no uniformity in the way how cell dies.

In addition, we investigated what characterizes the tendency of death in the presence of autotoxins. One possibility is the difference in cell cycle stages. We thus measured the death rate of the temperature-sensitive *cdc25-22* mutant strain, whose cell cycle can be arrested at the G2 phase at higher temperatures [23,24]. Using the *cdc25-22* mutant, we prepared synchronously dividing cells and found that they showed higher death rates than asynchronously dividing cells, which will contain a substantial portion of cells arrested in the G0 phase. This suggests that the cell cycle stage is one of the major factors that determine the cell competency to survive or die, and the resting cells might show tolerance to autotoxins (S18 Fig). Note that the above mutant cells producing fewer autotoxins show the similar tolerance to the autotoxins with WT cells (S19 Fig).

The identified toxins also facilitated cell adaptation in a condition-dependent manner. When we precultured cells in 3% MM with an inhibitory molecule, HICA or 2K3MVA, they grew in 0% MM with a sufficiently high concentration of inhibitory molecules that stopped the growth of nonadapted cells (Fig 3I and 3J). This indicated that the molecules we identified had 2 distinguishable effects: to kill latecomer cells and help firstcomer cells to adapt. Then, using RNAseq, we explored genes responsible for adaptation. We analyzed differentially expressed genes between 3% MM and 3% MM with 30 mM HICA or 25 mM 2K3MVA (S20 Fig). Gene ontology analysis revealed that many of the up-regulated genes in the adapted cells were involved in the transmembrane transport (exporters) and small molecule metabolic process (detoxification) (see Table D in S1 Text). We, therefore, constructed deletion mutants for 17 genes, which were included in the top 20 up-regulated genes in response to HICA (see Table E in S1 Text for a list of up-regulated genes). They were precultured in 3% MM with 30 mM HICA for the adaptation to autotoxins, and their growth was measured in 0% MM with

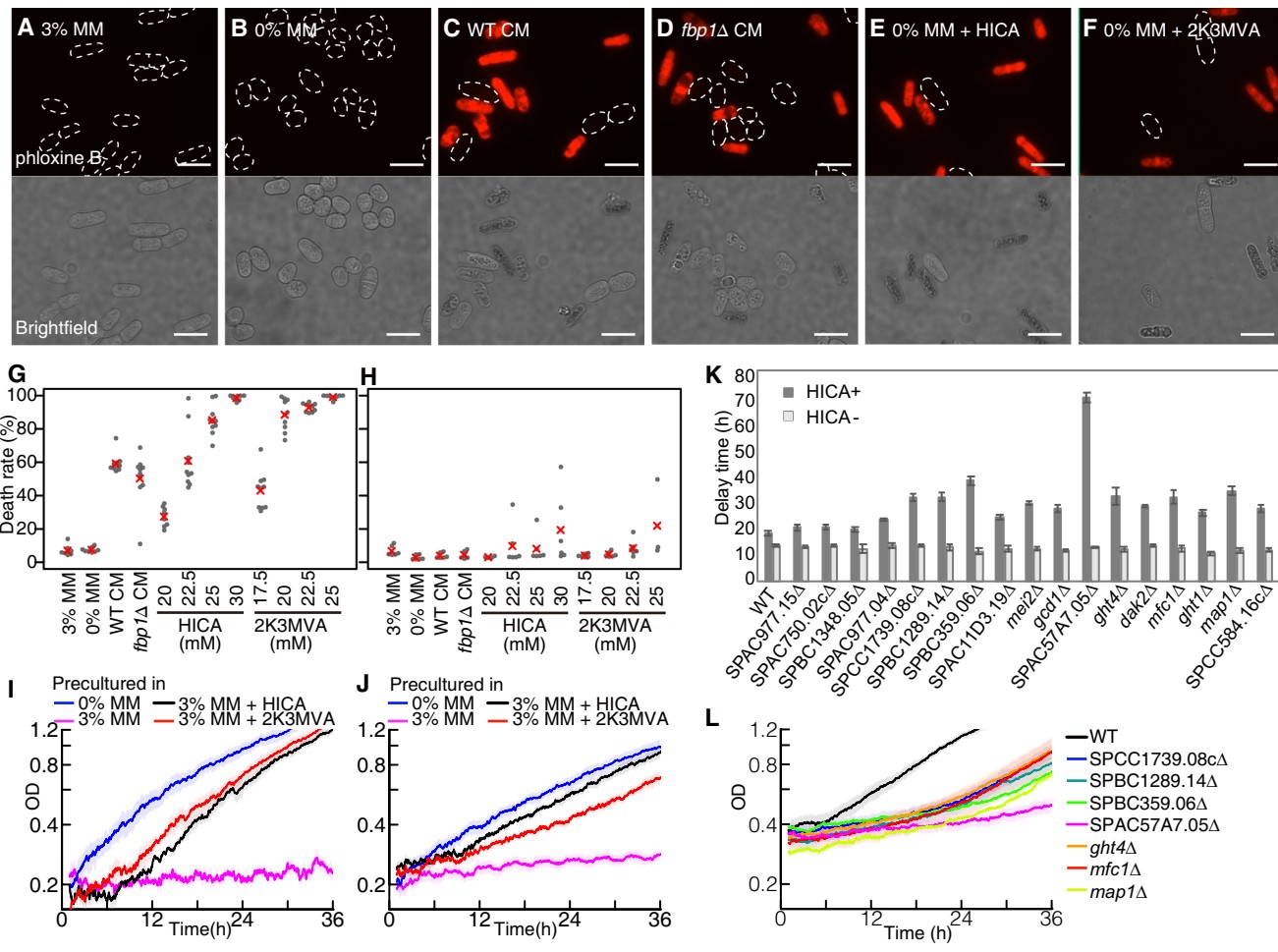

**Fig 3. Identified molecules kill cells and also facilitate cell adaptation to the molecule and deletion of some genes up-regulated in adapted cells prolonged the delay phase.** (A–F) Fluorescent (upper) and brightfield (bottom) microscopic images of WT cells in various media after 24 h of incubation. Cells precultured in 3% MM were transferred to (A) 3% MM, (B) 0% MM, (C) WT CM, (D) *fbp1*Δ CM, (E) 0% MM with 25 mM HICA, and (F) 22.5 mM 2K3MVA. In fluorescent microscopic images, dead cells were stained with phloxine B. Scale bar indicates 10 μm. (G and H) The dyed cell ratio after 8 h of incubation. Cells were precultured in (G) 3% MM (*n* = 8–10) or (H) 0% MM (*n* = 3–6). Grey dots represent the dyed cell ratio in each sample, and red crosses represent the mean value. (I and J) Growth curves of cells precultured in the presence of one of the inhibitory molecules along with 3% glucose, in 0% MM with (I) 30 mM HICA or (J) 25 mM 2K3MVA. Cells were precultured in 0% MM (blue line), 3% MM (pink line), 3% MM with 30 mM HICA (black line), and 3% MM with 25 mM 2K3MVA (red line). (K) Calculated length of the delay phase in deletion mutant strains, where the relevant genes were up-regulated in adapted cells (see Table E in S1 Text for a list of the genes). The deletion mutants and WT cells were precultured in 3% MM with 30 mM HICA for 24 h and inoculated into 0% MM with or without 30 mM HICA. The time point when the initial concentration doubled, τ, for the SPAC57A7.05Δ strain was much longer than the observed time range, and thus we calculated the time when the initial concentration was 1.5 times higher. Then, we extrapolated by multiplying the value by log2/log1.5. (L) Growth curves of some deletion mutants that showed the significant prolongation of the delay phase longer than 30 h in 0% MM with 30 mM HICA. Each line represents an average of *n*≥4 samples. See S13 Fig for the growth curves of the rest of the mutants. The data underlying this figure can be found in S1 and S3 Data. CM, conditioned media; MM, minimal media; WT, wild-type.

30 mM HICA. Notably, we detected substantial delay of adaptation in many of the deletion mutants (Fig 3K). In particular, the growth of SPCC1739.08cΔ, SPBC1289.14Δ, SPBC359.06Δ, SPAC57A7.05Δ, *ght4*Δ, *mfc1*Δ, and *map1*Δ strains was drastically suppressed, while that of those in 0% MM was normal (Figs 3K, 3L and S21). Those genes are predicted to encode a short-chain dehydrogenase, adducin, adducin, transmembrane transporter, plasma membrane hexose:proton symporter, prospore membrane copper transmembrane transporter, and DNA-binding transcription factor, respectively. These results suggest that cells undergo adaptation

by the mechanisms of both evacuation and detoxification to autotoxins. Those genes were also up-regulated in the WT CM with glucose and might contribute to the rescue of the cell growth by sugars (S22 Fig). In addition, those genes were up-regulated in survived cells in the WT CM or in the presence of the autotoxins (S23 Fig). At the same time, the lack of those genes did not affect the growth in 0% MM (S24 Fig). These results suggest that the adaptation to glucose depletion and autotoxins depends on several different processes.

Latecomer killing is not a unique characteristic of *S. pombe* but is widely observed in unicellular fungi. We cultured 2 strains of budding yeast *Saccharomyces cerevisiae*, phylogenetically distant from *S. pombe*, and prepared CM using them. Then, we found that such CM also initiated the delay phase in the growth of media producers (Figs 4A and S25). Moreover, we detected the same toxic molecules, HICA and 2K3MVA, in the media conditioned with *S. cerevisiae* (see Table A in S1 Text). In addition, the administration of such toxins to MM without glucose initiated the delay phase in a concentration-dependent manner (Figs 4B, 4C, S14B and S14C). Indeed, we confirmed that *S. cerevisiae* cells also died with the autotoxins similar to *S. pombe* (S26 Fig). Cells precultured under the glucose-depletion condition did not show a delay phase in their CM or 0% MM with an inhibitory molecule (Figs 4D, 4E S25D and S25E). This suggested that the same behavior with the same molecules, as observed in *S. pombe*, was evolutionarily conserved among distant species. In addition, media conditioned with distant species also initiated the delay phase (Fig 4F–4H), i.e., media conditioned with *S. pombe* inhibited the growth of *S. cerevisiae* and vice versa. Therefore, such a behavior was universally effective from closer to distant species.

## Discussion

In this paper, we reported a new ecological behavior for microbes; the latecomer killing. It seems similar to the toxin-antitoxin system, such as bacteriocins in bacteria [25,26] and killer factors in yeast [27,28] and paramecium [29], but it is fundamentally different. Although in the toxin-antitoxin system, the clonal cells are not killed [30], even the clonal latecomer cells can be killed when they have not been adapted to toxins in the mechanism discovered here. Such a behavior might help the yeast to avoid a mass suicide of the entire cell population and select an appropriate offspring that produces toxins and selfishly purify their genome from closely related species. Moreover, the latecomer killing may overcome the problems of the toxin-antitoxin system. In such a system, toxin producers should continuously produce antitoxins to protect themselves, and the maintenance of this state is a heavy burden for them [31]. Thus, the toxin producer is lost to a cheater, which only has the antitoxin system, whereas the cheater loses to cells with neither toxins nor immunity [32]. In contrast, the latecomer killing does not cost much because the adaptation system can be turned off without the toxin; indeed, the cells precultured in 3% MM without the autotoxins died in the CMs (Fig 1). In contrast, cheaters have disadvantages in their stable expression of protection mechanisms. Then, when the environment fluctuates between glucose-rich and glucose-depleted conditions like real natural conditions, cells with adaptation can beat cheaters in a glucose-rich situation, and cheaters would be expelled eventually. Thus, latecomer killing may be resistant to cheaters.

We found that distant yeast species universally conserved latecomer killing, even at the molecular level. This might be because the toxins in the reported system are simple molecules, while toxins in the toxin-immunity systems in bacteria and yeast are highly evolved proteins. In the bacteriocin system of *Escherichia coli*, toxins and receptors set off an arms race between the diversification of toxins and enhancement of their recognition by modifying the structure of proteins [33]. One of the reasons may be that the toxin-antitoxin system is vulnerable to the

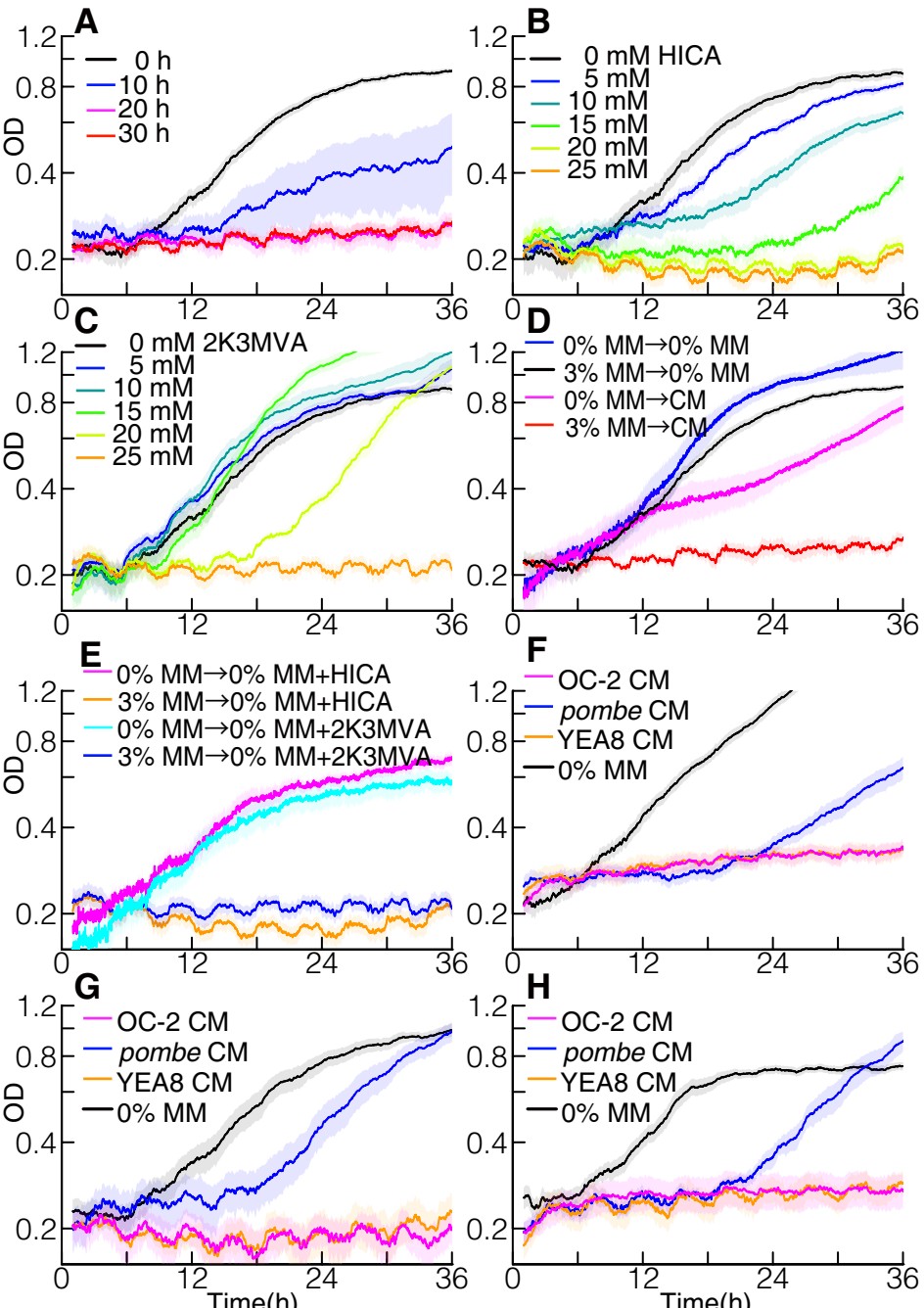

**Fig 4. Media conditioned with various strains of yeasts initiated the delay phase.** (**A**) Growth curves of *S. cerevisiae* (OC-2) cells in media conditioned with themselves. Different colored lines indicate growth curves of CM at different incubation times. Each line represents an average of $n \geq 6$ samples. (**B**) Growth curves of OC-2 in 0% MM with various concentrations of HICA. Each line represents an average of $n \geq 5$ samples. (**C**) Growth curves of OC-2 in 0% MM with various concentrations of 2K3MVA. Each line represents an average of $n \geq 5$ samples. (**D**) Growth curves of OC-2 precultured in 0% or 3% MM grown in OC-2 CM. Each line represents an average of $n \geq 6$ samples. (**E**) Growth curves of OC-2 precultured in 0% or 3% MM grown in 0% MM with 25 mM HICA or 25 mM 2K3MVA. Each line represents an average of $n \geq 6$ samples. (**F**, **G**, and **H**) Growth curves of (**F**) *S. pombe*, (**G**) OC-2, and (**H**) YEA8 in media conditioned with *S. pombe*, OC-2, or YEA8 for 30 h. Each line represents an average of $n \geq 6$ samples. The data underlying this figure can be found in S1 Data. CM, conditioned media; MM, minimal media.

invasion of cheaters, as described above, and, therefore, the toxin-antitoxin system cannot be maintained without evolving various toxins. In contrast, in the latecomer killing, the targets of toxins are insensitive to the detailed structure of molecules and are not specific receptors. Indeed, the enantiomers of HICA have the same activity, where both D- and L-forms of HICA cause the delay phase at the same concentration (S10 Fig). Moreover, HICA was first identified from fermented products of a bacterium, *Lactobacillus plantarum* [34]. HICA is reportedly useful for human health and food production: HICA is toxic to various bacterial species and pathogenic biofilms produced by *Candida* and *Aspergillus* species [35]. It is also produced by various lactic acid bacteria during ripening of Korean pickles, kimchi. The above facts suggest that HICA and 2K3MVA targets are universally conserved pathways. Therefore, the toxins we found were effective against a range of cells, from clones to distant species. Furthermore, such universality at the molecular level is consistent with the hypothesis discussed above that latecomer killing is resistant to the invasion of cheaters.

The latecomer killing we reported may play an important role in understanding the origin of multicellularity. Multicellular fungi fall into clades 8 to 11, while there are only 4 clades apart from fungi [36,37], indicating that transitions from unicellular to multicellular and multicellular to unicellular organisms occur easily in fungi. To form a complex multicellular body, mutual activation of growth, as well as growth inhibition and programmed cell death pathway, are essential [38]. Thus, the evolutionary origin or vestige of both mutual activation and inhibition of growth should be observed even in unicellular fungi. Indeed, multiple species of unicellular fungi use the former for quorum sensing [39]. However, the latter has not been reported. The mechanism we found here meets the criteria required for growth inhibition for multicellularity [40,41], i.e., the toxins cause cell death depending on the cell state and smoothly diffuse from cell to cell. The cell cycle dependency of cell death (S18 Fig) and the apoptotic pathway activation in a large fraction of dead cells (S17 Fig) may also suggest a possible relationship between latecomer killing and multicellularity. A recent artificial evolution experiment demonstrated that multicellular "snowflake" yeast repeatedly evolved 15 times from unicellular *S. cerevisiae* and underwent apoptosis to keep the original size constant [42]. This suggests that the origin or vestige of multicellularity is embedded in the unicellular yeast. The relationship between intercellular communication in unicellular cells and multicellularity is key to solving the enigma of major transitions in evolution [43], and our study provides a significant milestone.

## Materials and methods

### Continuous measurement of growth

Continuous measurement of the OD of yeast cultures was performed using ODBox-C/ODMonitor systems (Taitec, Saitama, Japan), a noncontact trubidimeter for shaking cultures. This apparatus detects and monitors the rate of transmitted light (950 nm) of shaken liquids in test tubes by voltage sensors every minute. ODBox-C was set in an incubator (BR-43FL, Taitec) and all the OD measurements were performed with 5 ml liquid cultures in 16.5 mm diameter test tubes at shaking speed of 180 rpm at 30° tilt angle. Following the manufacturer's instruction, calibration for yeast culture was performed to convert the raw voltage data from ODMonitor into $OD_{600}$ OD at 600 nm ($OD_{600}$ was measured using iMark Microplate Absorbance Reader (Bio-Rad Laboratories, Hercules, CA, USA). For growth measurement, each test tubes with media without cells were shaken for more than 30 min to stabilize the aeration effect, then yeast cells were added and OD was monitored. Detailed delay phase and death rate analyses are described in Texts A and D in S1 Text.

## Yeast strains and culture conditions

Haploid strains, L972 (h$^-$ 972) and AR110 (h$^-$ fbp1Δ), were used for the *S. pombe* experiments. Wine yeast OC-2 (IAM4274) and baker's yeast YEA8 were used for the *S. cerevisiae* experiments. Strain information is also listed in Table F in S1 Text. Initially, cells were precultured at 30˚C in yeast extract with supplements (YES; 3% Glucose, 0.5% yeast extract, 200 mg/L Adenine, 100 mg/L Uracil, 200 mg/L Histidine, 200 mg/L Leucine) liquid media. Then, cells were transferred to liquid MM with 3% glycerol and 3% glucose and precultured at 30˚C for about 24 h until $OD_{600}$ reached to 0.15 to 0.2 unless otherwise noted. The final concentration of MM was as follows: 14.7 mM potassium hydrogen phthalate, 15.5 mM $Na_2HPO_4$, 93.5 mM $NH_4CL$, 0.05% EtOH, 5.2 mM $MgCL_2$, 0.1 mM $CaCl_2 \cdot H_2O$, 13.4 mM KCl, 0.28 mM $Na_2SO_4$, 4.2 $\mu$M pantothenic acid, 81.2 $\mu$M nicotinic acid, 55.5 $\mu$M inositol, 40.8 nM D-Biotin, 4.76 $\mu$M citric acid, 8.09 $\mu$M boric acid, 2.37 $\mu$M $MnSO_4$, 1.39 $\mu$M $ZnSPO_4$ $7H_2O$, 0.74 $\mu$M $FeCl_3$ $6H_2O$, 0.81 $\mu$M $(NH_4)_6Mo_7O_{24} \cdot 4H_2O$, 0.60 $\mu$M potassium iodide, 0.16 $\mu$M $CuSO_4 \cdot 5H_2O$ 200 mg/L adenine, 100 mg/L uracil, 200 mg/L histidine, 200 mg/L leucine. The culture was then inoculated into various media including MM with 3% glycerol without glucose (0% MM) and CM. Before cells were transferred, they were washed twice with the new media.

## Preparation of conditioned media

Cells were precultured in 3% MM approximately to $OD_{600}$ 0.1 to 0.2, and they are washed twice with 0% MM, transferred to the same volume of 0% MM, and incuvated for 30 h. The final $OD_{600}$ was around 0.4 to 1.0 after 30-h incubation for *S. pombe* CM of WT, gene deletion mutants in Figs 3L and S12, *S. cerevisiae* CM as they grew in 0% MM. fbp1Δ CM is also prepared at starting $OD_{600}$ approximately 0.1 to 0.2, but the final $OD_{600}$ is still 0.1 to 0.2 after 30-h incubation. Cells were pelleted down by brief centrifugation (at room temperature, 3,000 · g for 1 min). The supernatants were then filtered using a 0.22-$\mu$m PVDF membrane (Millex Sterile Filter Unit, Merck Millipore, Burlington, MA, USA).

## Flow cytometry analysis for competition assays

Cells were genetically tagged with mNeonGreen (HN98) or mCherry (HN101). Strain information is also listed in Table F in S1 Text. They were precultured in 3% or 0% MM and transferred to 0% MM and WT CM.

mCherry and mNeonGreen fluorescence were detected at 561 nm and 488 nm and collected with 615/20 and 530/30 emission filters of NovoCyte flow cytometer (ACEA Biosciences, San Diego, CA, USA), respectively. mNeonGreen-tagged strain was constructed from pFA6a-mNeonGreen-HIS3MX6, which was a gift from Wei-Lih Lee (Addgene plasmid #129100; http://n2t.net/addgene:129100; RRID:Addgene_129100) The gating strategy for competition assay is shown in S7 Fig.

## Capillary electrophoresis time-of-flight mass spectrometry

Media composition was measured using an Agilent Capillary Electrophoresis System (CE-MS), and data were processed by Human Metabolome Technologies (Turuoka, Japan) in anion and cation analysis modes according to a previously published method [44]. Peaks with signal/noise ratios of more than 3-fold were extracted using MasterHands software ver2.17.1.11 (Keio University, Turuoka, Japan), and HMT's metabolite libraries were mined, including more than 900 metabolites to search for metabolites in the media.

## Search for growth inhibitors

Capillary electrophoresis-time-of-flight mass spectrometry (CE-MS) was performed in 2 biological replicates for 0% MM, and $fbp1\Delta$ CM, and in 3 biological replicates for WT CM. We extracted compounds found in both samples (compounds found in only 1 sample are described as N.R. in Table A in S1 Text), and 12 candidates of growth inhibitors were identified following the criteria described in the main text. Every detected compound was added to freshly prepared 0% and 3% MM at a final concentration of 10 to 40 mM, which was sufficiently high to certainly identify the candidates that are effective with a single dose. Then, the growth rate of cells in such media was measured using ODBox-C/ODMonitor systems. Purified reagents of the detected compounds were purchased and used for the experiments. Their information is as follows: L-Leucic Acid (Cat#L0026, Tokyo Chemical Industry (TCI), Tokyo, Japan; CAS RN: 13748-90-8), D-Leucic Acid (D—Hydroxyisocaproic acid; Cat#4026246, Bachem AG, Bubendorf, Switzerland; CAS RN: 20312-37-2), 3K2MVA (3-Methyl-2-oxovaleric Acid, Cat#K0020, TCI; CAS RN: 1460-34-0), 5-Oxoproline (DL-Pyroglutamic Acid; Cat#G0061, TCI; CAS RN: 149-87-1), 2-Hydroxyglutaric acid (Cat#H942575, Fujifilm Wako Pure Chemical Industries (Wako), Japan; CAS RN: 40951-21-1), Ala (L-Alanine; Cat#018–01043, Wako; CAS RN: 56-41-7), Gln (L(+)-Glutamine; Cat#076–00521, Wako; CAS RN: 56-85-9), Glu (L-Glutamic Acid; Cat#070–00502, Wako; CAS RN: 56-86-0), Glycerol 3-phosphate (Glycerophosphoric acid, aqueous solution 35% w/w; Cat#QA-1439, Combi-Blocks, San Diego, CA; CAS RN: 57-03-4), Hypoxanthine (Cat#H0311, TCI; CAS RN: 68-94-0), Inosine (Cat#099–00231, Wako; CAS RN: 58-63-9), Phe (L(-)-Phenylalanine; Cat#161–01302, Wako; CAS RN: 63-91-2), Succinic acid (Cat#190–04332, Wako; CAS RN: 110-15-6).

## HICA and 2K3MVA quantification by a liquid chromatography-mass spectrometry

The concentrations of HICA and 2K3MVA in various media were quantified using liquid chromatography-mass spectrometry (LC-MS/MS), and data were processed by Chemicals Evaluation and Research Institute (CERI), Japan. Measurements were performed on a Shimadzu Nexera XR liquid chromatography coupled to a Sciex QTRAP 5500 mass spectrometer with L-columun2 ODS Metal-free column (150 mm × 2.0 mm, 3 m; CERI).

## RNA sequencing and gene expression analysis by RNA sequencing

For gene expression analysis, 2 biological replicates were measured for each sample. As described above, 15 to 20 mL cultures 24 h after media change were prepared. Cells were pelleted down and frozen in liquid nitrogen. Frozen cell pellets were resuspended in 250 $\mu$L beads buffer (75 mM $NH_4OAc$, 10 mM EDTA, pH 8.0) at 65°C with 200 $\mu$L acid-washed glass beads (Sigma), 25 $\mu$L of 10% SDS, and 300 $\mu$L of acid-phenol:chloroform (pH 4.5, Thermo Fisher Scientific). The samples were vortexed 3 times 1 min each at 1-min intervals and incubated at 65°C, followed by a 10-min incubation at 65°C, 1-min vortexing, and 15-min centrifugation at room temperature (16,000 × $g$). Then, the upper aqueous phase was transferred to a fresh tube with 200 $\mu$L of beads buffer and 400 $\mu$L of phenol/chloroform/isoamyl alcohol (25:24:1, Sigma). Tubes were vortexed briefly and centrifuged at 4°C, 16,000 × $g$ for 15 min. The upper aqueous phase was transferred to a fresh tube with 600 $\mu$L ice-cold isopropanol and 21 $\mu$L 7.5 M $NH_4OAc$. Tubes were vortexed briefly and centrifuged at 4°C, 16,000 × $g$ for 30 min. After removing the supernatant, the pellet was washed with 70% ethanol, air-dried, and resuspended in nuclease-free water. RNA samples were treated with RQ1 DNase (Promega), followed by a ribosome RNA removal by the Ribo-Zero Gold rRNA Removal kit for yeast (Illumina).

cDNA libraries were prepared using NEBNext Ultra II Directional RNA Library Prep Kit for Illumina (New England Biolabs) following the manufacturer's instructions. cDNA libraries were sequenced using the Illumina paired-end technology on HiSeq (300-cycle). Two biological replicates were prepared for each sample. Salmon (version 1.1.0; [45]) was used to quantify RNA sequencing data. The reference genome and gene annotation list of *S. pombe* (version 2017.10.31) on the PomBase database [46] were used. For differential expression analysis, the R package "DESeq2" was used [47]. Likelihood ratio tests were performed to compare cells in 3% MM versus cells in 3% MM with HICA or 2K3MVA. Gene ontology enrichment analysis was performed for 464 genes that were significantly up-regulated or down-regulated ($p$-value < 0.01) in both 3% MM with 30 mM HICA and that with 25 mM 2k3MVA compared to 3% MM using GO Term Finder [48]. Sequences were deposited in DDBJ with DRA BioProject Acession Number: PRJDB11860.

## Gene expression analysis using reverse transcriptase polymerase chain reaction

RNA was purified as described above. For reverse transcription, PrimeScriptRT Reagent Kit with gDNA eraser (Takara Bio, Japan) was used following the manufacturer's instructions. The cDNA libraries were quantified using StepOne Real-Time PCR system (Thermo Fisher Scientific) with the Thunderbird SYBR qPCR Mix (Toyobo, Osaka, Japan) following the manufacturer's instructions. PCR primers used for qPCR are listed in Table G in S1 Text.

## Flow cytometry analysis to measure the dead cell ratio

Phloxine B (final concentration of 10 $\mu$g/ml, Sigma Aldrich) was used to dye dead cells. The fluorescent signal from phloxine B was detected at a wavelength of 488 nm and collected using 695/40 and 586/20 emission filters on a NovoCyte flow cytometer (ACEA Biosciences, San Diego, CA, USA). The gating strategy for the identification of dead cell population is shown in S16 Fig.

## Microscopy and imaging analysis

All images were captured using a microscope equipped with UPlanSApo 100×/1.40 Oil Objective (Olympus, Tokyo, Japan) on an EVOS FL Imaging System (Thermo Fisher Scientific, Waltham, MA, USA). Images were acquired using a SonyICX285AL monochrome CCD camera controlled with built-in software for image acquisition. Fluorescence signal from phloxine B was detected at a wavelength of 530 nm and collected by EVOS Light Cube (Texas Red), which contains a 628/32 emission filter (Thermo Fisher Scientific).

## Costaining of annexin V-FITC and propidium iodide

AnnV/PI staining was performed using the ApoAlert Annexin V-FITC Apoptosis Kit (Clontech) as previously described [49–51]. Approximately $1 \times 10^7$ cells were first harvested and washed with Sorbitol buffer (1.2 M Sorbitol, 0.5 mM MgCl$_2$, and 35 mM potassium phosphate, pH 6.8), and then treated with 10 mg/ml Zymolyase-20T (Nacalai) for 2 h at room temperature. Spheroplasts were then washed and resuspended in 1 × Binding Buffer, followed by addition of 5 $\mu$l of Annexin V-FITC and 10 $\mu$l of PI according to the manufacturer's instructions. After 15-min incubation at room temperature in the dark, the samples were analyzed by a flow cytometry (NovoCyte) using a single laser emitting excitation light at 488 nm.

## Death rate analysis of synchronous culture of *cdc25-22*

Synchronous cultures were prepared by transient temperature shifts using *cdc25-22*, temperature-sensitive mutant as previously reported [23,24,52]. Cells were precultured in 15.5 ml of 3% MM at 25˚C for 24 h to $5 \times 10^5$ cells/ml. Then, the cells were shifted to 36˚C for 4.25 h followed by shifting the temperature to the permissive temperature of 25˚C to release the G2 block. The cells were kept culturing at 25˚C in the 3% MM for 0 h (G2 phase), 1 h (M/G1 phase), and 2 h (S phase), and 450 $\mu$l of each culture were sampled for PI staining for cell synchronization analysis. Asynchronous cells were prepared without 36˚C heat shock. The cells in the rest of the 15 ml cultures in each phase were dispensed into 3 aliquots, washed, and pelleted down followed by resuspension to 1 ml of 0% MM with 20, 22.5, or 25 mM HICA and 10 $\mu$g/ml phloxine B. After 8 h from the media change, the dead cell ratio in each aliquot was measured by a flow cytometer as described above. Cells were fixed with 50 $\mu$l formalin for 10 min on ice, washed twice in PBS, and treated with 0.2 mg/ml RNaseA in PBS/50mM EDTA for 2 h at 37˚C. Then, PI was added to the final concentration of 5 $\mu$g/$\mu$l, detected by a flow cytometer as described above.

## Measurement of surviving cell growth in conditioned media

WT cells were cultured in WT CM for 24 h, and the culture was spread on YES agar plates to isolate single colonies. After incubating at 30˚C for 3 days, 24 colonies were picked and cultured in YES liquid medium for 24 h and then in 3% MM for 24 h. Then, the cells were cultured in WT CM again, and their growth curves were constructed using ODBox-C/ODMonitor systems.

## Genome sequencing of surviving cells in conditioned media

WT cells were cultured in WT CM with phloxine B (final concentration of 10 $\mu$g/ml) for 24 h. Unstained living cells were selected using a cell sorter SH-800 (SONY) using a 488-nm excitation wavelength laser with a 525/50 emission filter and 561 nm excitation wavelength laser with a 617/30 emission filter. Then, they were cultured in YES liquid medium at 30˚C for 3 days and pelleted down to extract genomic DNA. The cell pellet was resuspended in 150 $\mu$l of STES Buffer (0.5 M NaCl, 0.01 M EDTA, 1% SDS) with 150 $\mu$l of glass beads and 150 $\mu$l of phenol-chloroform. The samples were vortexed for 10 min, followed by a 10-min centrifugation at room temperature, 16,000 $\times$ g. The upper aqueous phase was transferred to a fresh tube with 150 $\mu$l of phenol-chloroform. The tubes were vortexed, followed by a 10-min centrifugation at 16,000 $\times$ g at room temperature. The upper aqueous phase was transferred to a fresh tube and mixed with 15 $\mu$l of 3 M NaOAc and 375 $\mu$l of ethanol. The samples were mixed briefly and centrifuged for 10 min at room temperature (16,000 $\times$ g). After the supernatant was removed, the pellet was rinsed with 70% ethanol, air-dried, and resuspended in nuclease-free water.

Genomic DNA libraries were prepared using the NEBNext Ultra II DNA Library Prep Kit for Illumina (New England Biolabs, Ipswich, MA, USA), following the manufacturer's instructions. Genomic DNA libraries were sequenced using the Illumina paired-end technology on MiSeq with MiSeq Reagent Kit v3 (600-cycle; Illumina, San Diego, CA, USA). Detailed information is provided in Text B in S1 Text. Sequences were deposited in DDBJ with DRA BioProject Acession Number: PRJDB10422.

## Supporting information

**S1 Fig. Glycerol is essential for growth in the 0% MM.** Growth curves of WT cells in the MM without glucose and with 3% glycerol (black) and that without both glucose and glycerol (red).

Cells were precultured in the MM with 3% glucose and 3% glycerol. Each line represents an average of *n* = 2 samples. The data underlying this figure can be found in S2 Data.
(PDF)

**S2 Fig. Effect of auxotrophic marker supplements; leucine, adenine, uracil, and histidine are not essential for the delay phase.** Growth curves of WT cells in the MM with and without leucine, adenine, uracil, and histidine, and those media conditioned by the WT cell. Each line represents an average of *n* = 4–8 samples. The data underlying this figure can be found in S2 Data.
(PDF)

**S3 Fig. Growth curve of *fbp1Δ* without glucose.** *fbp1Δ* and WT cells were precultured in 3% MM and then transferred to 0% or 3% MM. Growth curves of *fbp1Δ* cells in 0% and 3% MM are shown in blue and light blue, respectively, and those of WT cells are shown in black and red, respectively. Each line represents an average of 3–7 samples. The data underlying this figure can be found in S2 Data.
(PDF)

**S4 Fig. Delay in growth in the CM was rescued by sugars but not by glutamate or amino acid mix.** (**A**) Growth curves of WT cells in the WT CM with 3% glucose, 3% fructose, 3% galactose, 3% mannose, or 3% 2-deoxy-D-glucose (2-DG). Each line represents an average of *n* = 4–6 samples. (**B**) Growth curves of WT cells in the WT CM with 3.75 g/L (22.2 mM) monosodium glutamate or an amino acid mix. The final amino acid concentration in amino acid mix sample was as follows: Adenine 10 mg/L, L-Arginine HCl 50 mg/L, L-Aspartic Acid 80 mg/L, L-Histidine HCl 20 mg/L, L-Isoleucine 50 mg/L, L-Lysine HCl 50 mg/L, L-Methionine 20 mg/L, L-Phenylalanine 50 mg/L, L-Threonine 100 mg/L, L-Tryptophan 50 mg/L, Uracil 20 mg/L, L-Tyrosine 50 mg/L, L-Valine 140 mg/L. Each line represents an average of *n* = 2–4 samples. The data underlying this figure can be found in S2 Data.
(PDF)

**S5 Fig. Growth curves of the cells that survived in WT CM.** The blue line shows the average growth curves of cells in 24 independent colonies isolated from surviving cells in WT CM. Red and magenta lines are the average growth curves of original WT cells precultured in 3% and 0% MM, respectively. The data underlying this figure can be found in S2 Data.
(PDF)

**S6 Fig. Histograms of quality scores calculated using GATK for each mutation in original WT cells and cells that survived in WT CM.** The common mutations detected in all 4 samples are shown in yellow. Mutations detected in both original WT and surviving cells, but not in all 4, are shown in red. Unique mutations detected only in a certain replicate of the original WT, and surviving cells are shown in blue and cyan, respectively. The data underlying this figure can be found in S6 Data.
(PDF)

**S7 Fig. Flow cytometry gating strategy for identification of the competition assay discussed in Fig 1H and 1I.** (**A-D**) Fission yeast cells were first gated on the red E1 gate (left panel). Then, cells in the E2 area were defined as mCherry-positive cells and cells in the E3 area were defined as mNeon-Green-positive cells (right panel). The total count of each sample is 500,000 counts. (**A**) An example image of competition assay between mCherry-tagged cells precultured in 0% MM and mNeonGreen-tagged cells precultured in 3% MM at 0 h. (**B**) An example image of competition assay between mCherry-tagged cells precultured in 0% MM and mNeonGreen-tagged cells precultured in 3% MM at 72 h. (**C**) An example image of

competition assay between mCherry-tagged cells precultured in 0% MM and mNeonGreen-tagged cells precultured in 0% MM at 0 h. (**D**) An example image of competition assay between mCherry-tagged cells precultured in 0% MM and mNeonGreen-tagged cells precultured in 0% MM at 72 h.
(PDF)

**S8 Fig. Release of inhibitory compounds and cell adaptation to them is beneficial for the competition among clonal cells.** mNeonGreen- and mCherry-labelled WT cells were mixed in equal fractions in the WT CM at 0 h, and then they showed population dynamics. Green and red areas indicate the fraction of mNeonGreen- and mCherry-labelled cells, respectively, and overwriting outline characters indicate preculture conditions, i.e., 3% and 0% indicate cells precultured in 3% and 0% MM, respectively. Black vertical bars between 2 areas indicate SEM (number of each sample is 12). (**A**) Competition assay between mNeonGreen-labelled cells precultured in 0% MM and mCherry-labelled cells precultured in 3% MM. (**B**) Competition assay between mNeonGreen- and mCherry-labelled cells precultured in 3% MM. The data underlying this figure can be found in S1 Data.
(PDF)

**S9 Fig. Growth curves of WT cells in the CM treated with autoclave, nucleases, or proteases.** WT cells were precultured in 3% MM, then growth was measured in the various CM treated as follows: Autoclaved: WT CM was autoclaved at 121˚C for 20 min. Papain treated: WT CM was treated with 2.5 mg/ml papain (Cat#166–00171, Wako) at 37˚C for 24 h. RNase A treatment: WT CM was treated with 50 $\mu$g/ml RNaseA (Cat#318–06391, Nippon gene) at 37˚C for 24 h. DNase treatment: WT CM was treated with 100 U/ml DNase I (Cat#18047019, Invitrogen) at 37˚C for 24 h. Proteinase K treatment: WT CM was treated with 8 $\mu$g/ml Proteinase K (Cat# 25530049, Invitrogen) at 37˚C for 24 h. Each line represents an average of $n$ = 2–4 samples. The data underlying this figure can be found in S2 Data.
(PDF)

**S10 Fig. Growth curves of WT cells in 0% MM with enantiomers of HICA.** (**A** and **B**) Structures of (**A**) L- and (**B**) D-forms of HICA. (**C** and **D**) Growth curves of WT cells in 0% MM with various concentrations of (**C**) L-form HICA and (**D**) D-form HICA. For the L-form of HICA, growth curves are an average of 15–18 samples, and for the D-form of HICA, it is an average of 2 samples. The data underlying this figure can be found in S2 Data.
(PDF)

**S11 Fig. Adding of BCAAs to the 0% MM did not inhibit the growth.** Growth curves of WT cells in 0% MM without auxotrophic marker supplements with 30 mM, 60 mM leucine or mixture of branched chain amino acids (BCAA: 30 mM leucine + 30 mM valine + 30 mM isoleucine). WT cells were precultured in 3% MM without auxotrophic marker supplements. Each line represents an average of $n$ = 2 samples. The data underlying this figure can be found in S2 Data.
(PDF)

**S12 Fig. Growth curves of WT cells in media conditioned by deletion mutant strains of genes encoding putative hydroxyacid dehydrogenase.** Different colored lines indicate growth curves of WT in media conditioned by different deletion mutants. Each deleted gene is predicted as putative hydroxyacid dehydrogenase or hydroxyacid dehydrogenase homolog in the UniProt database. WT cells were precultured in 3% MM. Each line represents an average of $n$ = 4–7 samples. The data underlying this figure can be found in S2 Data.
(PDF)

**S13 Fig. Add-back experiments of HICA and 2K3MVA into conditioned media from deletion mutants for hydroxyacid dehydrogenases.** Cells precultured in 3% MM were inoculated to media conditioned by (**A**) SPACUNK4.10Δ, (**B**) SPBC1773.17cΔ, (**C**) SPCC364. 07Δ, (**D**) SPAC186.07cΔ, and (**E**) SPAC186.02Δ mutants. The conditioned media from those cultures were added with HICA or 2K3MVA or both of them. (**F**) Length of the delay phase $\tau$ for each growth curve. A line for each conditioned media represents an average of $n$ = 4–7 samples, and those for conditioned media with the autotoxins represents an average of $n$ = 2. The data underlying this figure can be found in S2 Data.
(PDF)

**S14 Fig. Schematic representation of growth curves with delay phase.** $a_0$ is the OD at time = 0, $\tau$ is the time when the OD reaches twice of $a_0$, and $r$ is the growth rate at the steady growth phase. The blue dotted line and orange dashed line are the concentrations of dead and living cells, respectively; thus, the solid magenta line, which is the summation of dead and living cells, is observed as OD.
(EPS)

**S15 Fig. Measured and estimated death rate in various media.** Open circles represent the death rate measured using phloxine B, as shown in Fig 3. Blue crosses represent the average of each sample. Filled circles represent the estimated death rate as described in Text B in S1 Text. Red crosses represent the average of each sample. The data underlying this figure can be found in S1 and S2 Data.
(EPS)

**S16 Fig. Flow cytometry gating strategy for identification of the dead cell populations discussed in Fig 3G and 3H.** (**A**-**D**) Fission yeast cells were first gated on a FSC/SSC scatter plot as the red P1 gate (left panel). Then, cells gated on the blue and yellow gates were defined as phloxine B-stained dead cells and unstained living cells, respectively (right panel). The total count of each sample is 100,000 counts. Example images of flow cytometry data for cells in (**A**) 3% MM, (**B**) 0% MM, (**C**) WT CM, and (**D**) 30 mM HICA.
(PDF)

**S17 Fig. Discrimination between early apoptosis and early necrosis by using costaining with annexin V (AnnV) and propidium iodide (PI).** (**A**) AnnV/PI staining of WT cells in various media. Cells were inoculated to 0% MM, WT CM, 0% MM with 20/25/30 mM HICA, and 0% MM with 15/20/25 mM 2K3MVA from 3% MM (0h), and then stained at 2 h and 4 h after the inoculation. The bottom left panel shows the schematic image of the gate setting and their biological annotations for the panel (**B**). Early apoptotic cells exhibit phosphatidylserine externalization, which is detected by AnnV staining (AnnV+/PI−). Primary necrotic cells show a ruptured plasma membrane, which is detected by PI staining (AnnV−/PI+). Late apoptotic/secondary necrotic cells show both phosphatidylserine externalization and membrane permeability (AnnV+/PI+), while living cells are not stained with both AnnV and PI (AnnV−/PI−). B) Proportion of the cells in each phase. Percentages of cells in the early apoptosis (blue), in the primary necrosis (yellow), in the late apoptosis/secondary necrotic phase (orange), and alive cells (grey) are shown. The data underlying this figure can be found in S2 Data.
(PDF)

**S18 Fig. Death rate of synchronous and asynchronous cells.** (**A**) Death rate of the *cdc25-22* strain with and without a transient heat shock in 0% MM with HICA. G2, M/G1, and S cells were transferred to 0% MM with HICA 0, 1, and 2 h after a transient increase in temperature

to 36˚C. Death rate was measured after 8 h from media change. Asynchronous cells were transferred without a heat shock. The number of biological replicates is $n$ = 2. (**B**) Flow cytometry analysis of synchronous cells. The first peak indicates the 1c cells, whose DNA amount is that of an interphase cell containing a G2 nucleus or a mitotic cell containing two G1 nuclei. The second peak indicates the 2c cells, whose DNA amount is that a septated cell containing 2 G2 nuclei. The ratio of 1c and 2c cells in each sample is given as: 87.8 ± 2.1%: 12.2 ± 2.1% for G2 cells, 76.0 ± 6.6%: 24.0 ± 6.6% for M/G1 cells, and 66.0 ± 1.7%: 34.0 ± 1.7% for S cells. The data underlying this figure can be found in S2 Data.
(PDF)

**S19 Fig. Growth curves of deletion mutant strains of genes encoding putative hydroxyacid dehydrogenase in 0% MM with HICA or 2K3MVA.** The strains in S12 Fig cells were precultured in 3% MM and then shifted to 0% MM, 0% MM with 25, 27.5 mM HICA or 20, 25 mM 2K3MVA. Each line represents an average of $n$ = 2–4 samples. The data underlying this figure can be found in S2 Data.
(PDF)

**S20 Fig. Genes up-regulated or down-regulated in the adapted cells.** MA-plot for the log fold change of all genes between cells cultured in 3% MM with 30 mM HICA vs. those in 3% MM (**A**), and between cells cultured in 3% MM with 25 mM 2K3MVA vs. those in 3% MM (**B**). Cells precultured in 3% MM were cultured 3% MM with 30 mM HICA, 3% MM with 25 mM 2k3MVAm, and 3% MM for 24 h. Then, a difference in the gene expression was analyzed by RNA-seq. Red points indicate genes with a $p$-value less than 0.01. The data underlying this figure can be found in S7 Data.
(PDF)

**S21 Fig. Growth curves of some deletion mutants that showed the prolongation of the delay phase less than 30 h in 0% MM with 30 mM HICA.** Each line represents an average of $n{\geq}4$ samples. See Fig 3L for growth curves of the mutants that showed the significant prolongation of the delay phase longer than 30 h. The data underlying this figure can be found in S1 Data.
(PDF)

**S22 Fig. Relative changes in gene expression in 3% MM with HICA/2K3MVA or in the WT CM with 3% glucose.** WT cells were precultured in 3% MM and inoculated to 3% MM with 30 mM HICA or 25 mM 2K3MVA and CM with 3% glucose. Expression of genes of which deletion mutants showed the significant prolongation of the delay phase, as shown in Fig 3L, was quantified by RT-PCR at times 0, 1, 4, 8, and 24 h after a change of media. Each line represents an average of $n$ = 2 samples. The data underlying this figure can be found in S2 Data.
(PDF)

**S23 Fig. Relative changes in gene expression of survived cells in the CM or media with the autotoxins.** WT cells were precultured in 3% MM and inoculated to WT MC, 0% MM with or without 10, 25 mM HICA, 10, or 20 mM 2K3MVA, and 3% MM with 30 mM HICA or 25 mM 2K3MVA. Expression of genes of which deletion mutants showed the significant prolongation of the delay phase, as shown in Fig 3L, was quantified by RNAseq at times 0, 8, and 24 h after a change of media. Each line represents an average of $n$ = 2 samples. The data underlying this figure can be found in S2 Data.
(PDF)

**S24 Fig. Growth curves of the deletion mutants of genes for the adaptation in the 0% MM.** Mutant cells that showed the significant prolongation of the delay phase in Fig 3L were

precultured in 0% MM, and their growth was measured in WT CM. Each line represents an average of $n = 2$ samples. The data underlying this figure can be found in S2 Data.
(PDF)

**S25 Fig. Media conditioned with various yeast strains caused the delay phase.** (**A**) Growth curves of 2 strains of *S. cerevisiae* (YEA8) in media conditioned by itself. Different colored lines indicate growth curves in CM with different incubation times. Each line represents an average of $n \geq 6$ samples. (**B**) Growth curves of YEA8 in 0% MM with various concentrations of HICA. Each line represents an average of $n \geq 5$ samples. (**C**) Growth curves of YEA8 in 0% MM with various concentrations of 2K3MVA. Each line represents an average of $n \geq 5$ samples. (**D**) Growth curves of YEA8 precultured in 0% MM or 3% MM in CM of OC-2. Each line represents an average of $n \geq 6$ samples. (**E**) Growth curves of YEA8 precultured in 0% MM or 3% MM in 0% MM with 25 mM HICA or 25 mM 2K3MVA. Each line represents an average of $n \geq 6$ samples. The data underlying this figure can be found in S2 Data.
(PDF)

**S26 Fig. The autotoxins also killed *S. cerevisiae* cells.** (**A** and **B**) Fluorescent (upper) and brightfield (bottom) microscopic images of WT *S. cerevisiae* strains (**A**) OC2, (**B**) YEA8 in various media after 8 h of incubation. Cells precultured in 3% MM were transferred to 3% MM, 0% MM, CM of each strain, 0% MM with 15 mM HICA, and 15 mM 2K3MVA. In fluorescent microscopic images, dead cells were stained with phloxine B. Scale bar indicates 10 $\mu$m. (**C** and **D**) Box plot of the phloxine B stained cell ratio after 8 h of incubation. *S. cerevisiae* (**C**) OC2, (**D**) YEA8 cells were precultured in 3% MM and shifted to various media. Grey areas represent the interquartile ranges of dyed cell ratio in each sample, and crosses represent the mean value. Fluorescence of over 50,000 cells were measured for each sample using FACS. ($n = 2$–4) The data underlying this figure can be found in S2, S4, and S5 Data.
(PDF)

**S1 Text. Text A-D and Table A-G.**
(PDF)

**S1 Data. The individual numeric values in Figs 1B, 1C, 1E, 1G–1I, 2D–2I, 3G–3L, 4A–4H, S8A, S8B and S21.**
(XLSX)

**S2 Data. The individual numeric values in S1–S5, S9–S13, S15–S19 and S22–S26 Figs.**
(XLSX)

**S3 Data. Raw microscopic images of Fig 3.**
(PS)

**S4 Data. Raw microscopic images of S26A Fig.**
(PS)

**S5 Data. Raw microscopic images of S26B Fig.**
(PS)

**S6 Data. Raw VCF files for S6 Fig.**
(ZIP)

**S7 Data. Raw differential genome expression data for S20 Fig.**
(ZIP)

## Acknowledgments

The authors would like to thank H. Nakaoka for his help in preparing *S. pombe* strains HN98 and HN101. The authors would like to thank C. Furusawa for fruitful research discussion, and E. Takaya for experimental supports. Yeast strain FY7755 was provided by the National Bio-Resource Project (NBRP), Japan.

## Author Contributions

**Conceptualization:** Arisa H. Oda, Tetsuhiro S. Hatakeyama.

**Data curation:** Arisa H. Oda, Miki Tamura, Tetsuhiro S. Hatakeyama.

**Formal analysis:** Arisa H. Oda, Tetsuhiro S. Hatakeyama.

**Funding acquisition:** Arisa H. Oda, Kunihiko Kaneko, Kunihiro Ohta, Tetsuhiro S. Hatakeyama.

**Investigation:** Arisa H. Oda, Miki Tamura, Kunihiro Ohta, Tetsuhiro S. Hatakeyama.

**Methodology:** Arisa H. Oda, Miki Tamura, Tetsuhiro S. Hatakeyama.

**Project administration:** Arisa H. Oda, Kunihiko Kaneko, Kunihiro Ohta, Tetsuhiro S. Hatakeyama.

**Resources:** Arisa H. Oda, Miki Tamura.

**Supervision:** Kunihiko Kaneko, Tetsuhiro S. Hatakeyama.

**Validation:** Arisa H. Oda, Kunihiko Kaneko, Kunihiro Ohta, Tetsuhiro S. Hatakeyama.

**Visualization:** Arisa H. Oda, Tetsuhiro S. Hatakeyama.

**Writing – original draft:** Arisa H. Oda, Tetsuhiro S. Hatakeyama.

**Writing – review & editing:** Arisa H. Oda, Kunihiko Kaneko, Kunihiro Ohta, Tetsuhiro S. Hatakeyama.

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
