## [Editor Report · Decision Letter 0]

22 Mar 2022

Dear Dr Hatakeyama, 

Thank you for submitting your manuscript entitled "Autotoxin-mediated voluntary triage in starved yeast community" for consideration as a Research Article by PLOS Biology.

Your manuscript has now been evaluated by the PLOS Biology editorial staff, as well as by an academic editor with relevant expertise, and I'm writing to let you know that we would like to send your submission out for external peer review.

Once your full submission is complete, your paper will undergo a series of checks in preparation for peer review. Once your manuscript has passed the checks it will be sent out for review. To provide the metadata for your submission, please Login to Editorial Manager (https://www.editorialmanager.com/pbiology) within two working days, i.e. by Mar 24 2022 11:59PM.

If your manuscript has been previously reviewed at another journal, PLOS Biology is willing to work with those reviews in order to avoid re-starting the process. Submission of the previous reviews is entirely optional and our ability to use them effectively will depend on the willingness of the previous journal to confirm the content of the reports and share the reviewer identities. Please note that we reserve the right to invite additional reviewers if we consider that additional/independent reviewers are needed, although we aim to avoid this as far as possible. In our experience, working with previous reviews does save time. 

If you would like to send previous reviewer reports to us, please email me at rroberts@plos.org to let me know, including the name of the previous journal and the manuscript ID the study was given, as well as attaching a point-by-point response to reviewers that details how you have or plan to address the reviewers' concerns. 

Given the disruptions resulting from the ongoing COVID-19 pandemic, please expect some delays in the editorial process. We apologise in advance for any inconvenience caused and will do our best to minimize impact as far as possible.

Kind regards,

Roli Roberts

Roland Roberts

Senior Editor

PLOS Biology

rroberts@plos.org

---

## [Decision Letter · Decision Letter 1]

6 May 2022

Dear Dr Hatakeyama,

Thank you for your patience while your manuscript "Autotoxin-mediated voluntary triage in starved yeast community" was peer-reviewed at PLOS Biology. It has now been evaluated by the PLOS Biology editors, an Academic Editor with relevant expertise, and by four independent reviewers. 

In light of the reviews, which you will find at the end of this email, we would like to invite you to revise the work to thoroughly address the reviewers' reports.

IMPORTANT: You'll see that all of the reviewers are intrigued by the phenomenon that you describe, but each of them has a significant number of requests, some involving further experimental work. We note that some of reviewer #1's requests, while they would strengthen our understanding of the implications of your observations, go somewhat beyond the scope of the current study. However, reviewer #1 also makes the interesting suggestion that your paper might work better as a Short Report; indeed, when invited to cross-comment on the other reviews, reviewer #3 also made this suggestion. We therefore give you two choices:

EITHER: Address all concerns, including all of those raised by reviewer #1, and resubmit your revised version as a full Research Article...

OR: Address all of the concerns of reviewers #2, #3 and #4, plus those of reviewer #1 that are needed to fully support the current claims, and resubmit your revised version as a Short Report. IMPORTANT: If you go down this route, you need to reduce the number of Figures to a maximum of 4 (either by combining the current Figures or moving material to the Supplement); if you cannot change the article type in our system, we can do so on your behalf.

Given the extent of revision needed, we cannot make a decision about publication until we have seen the revised manuscript and your response to the reviewers' comments. Your revised manuscript is likely to be sent for further evaluation by all or a subset of the reviewers.

**IMPORTANT - SUBMITTING YOUR REVISION**

*Re-submission Checklist*

*Published Peer Review*

*PLOS Data Policy*

*Blot and Gel Data Policy*

Sincerely,

Roli Roberts

Roland Roberts

Senior Editor

PLOS Biology

rroberts@plos.org

REVIEWERS' COMMENTS:

Reviewer #1:

In this study titled 'Autotoxin-mediated voluntary triage in starved yeast community', the authors identify a novel phenomenon wherein S. pombe cell populations adapt to glucose starvation by releasing autotoxins. The authors identify two compounds (HICA and 2K3MVA) that putatively act as autotoxins driving this collective cellular phenomenon. Only a fraction of the cells in the population survive as these autotoxins build up. Interestingly, this phenomenon seems to be evolutionarily conserved, even in the distantly related S. cerevisiae cells, suggesting some universality.

This study reports a novel, interesting phenomenon. However, this study definitely overstates many claims, or makes plausible inferences without sufficient data.

Upfront, the highlight of the study is the discovery of this collective phenomenon mediated by produced metabolites that are accessible to all cells, but leading to some cells dying and others selectively surviving. Three major points are unclear: (i) whether such a phenomenon provides any advantage to cells within the population (and what it might mean to a population of clonal cells), (ii) while there is no molecular/mechanistic insight (and let me state that there need not always be molecular/mechanistic insight) into how this phenomenon might be regulated, it is not clear even from the data presented whether the phenomenon is purely a reproducible, stochastic event, or if there are any deterministic features that would determine which cells might be resistant to auto-toxins, or might benefit from these molecules, and (iii) is there a separation within the population into cells that produce the toxins vs others that do not produce toxins/are resistant to toxins/consume toxins. In may ways, this study works better as a 'short report' that highlights this phenomenon, rather than a full article that does not provide mechanistic or conceptual insight into how such a phenomenon might occur.

The following are the major concerns with this study.

Major concerns

1. The authors suggest that the presence of inhibitory compounds and not nutrient depletion is responsible for the delay in growth observed in cells in CM. They show that glucose addition rescues this delay in growth suggesting that the inhibitory effect is absent when glucose is present (fig 2A). However, this experiment does not rule out the possibility that glucose-depletion itself is responsible for the delay in growth after shift to CM. The fbp deletion experiment does not answer this directly. What happens if an alternate 'good' carbon source (eg. glycerol) or an amino acid mix/glutamate is added to the CM? Will the delay in growth be rescued? This experiment will tell us whether the growth delay is independent of all nutrient availability or not. It will also strengthen the observation that the rescue in growth is glucose specific.

2. It is not clear from the data at what point in time do cells in 0% MM start producing HICA and 2K3MVA. Are these present only in CM or do cells which are shifted to CM continue to produce these compounds? Can the amount of these compounds be measured over time to address this?

3. Relatedly, is there a build up and then a decrease in these metabolites? If the compounds only accumulate, then the interpretation is different from if the metabolites accumulate and then start decreasing (suggesting some utilization and/or breakdown), which holds even if the metabolites in the medium flat-line. An additional possibility is that there is a feeder-utilizer type phenomenon that is established, where some cells continue to put out these metabolites, while others do not produce it (and possibly also start utilizing these metabolites). Can the authors establish which of these possibilities exist? If so, this becomes conceptually very interesting (from the perspective of collective cell behavior), regardless of which process occurs.

4. Relatedly, the authors can definitely identify cells that now are able to grow in medium which as these autoinhibitors. Can these cells actually start utilizing/breaking down these metabolites? A possible experiment would be to separate the cells that are now growing in this medium, and then add conditioned medium (exogenously) to these cells, and see if these metabolites are taken up and consumed. This experiment would be quite important in order to understand what is happening with this autoinhibitory system (and interpretations therin).

5. (points 5-7 are related queries) Do the authors have any idea how these compounds (eg. HICA) are produced in cells? Eg. is there any evidence of high branched-chain amino acid catabolism or keto-genesis, or is this just a reflection of an accumulation of BCAAs (which are all ketogenic)? Can, for example, just the addition of leucine/BCAAs after glucose depletion result in this phenomenon of cells dying? 

6. Can the production of these compounds be disrupted genetically? Will these mutant cells behave differently from the WT cells in terms of glucose starvation response?

7. Relatedly, in complementary experiments, if the authors keep replacing the medium with glucose depleted medium without the HICA and the keto-acids, will cells in the population now start dying/growing in a similar manner? This relates to the idea of whether it is actually HICA that causes cell death, or is it just the accumulation of leucine that is sufficient for this phenomenon. Leucine is a ketogenic acid, and also can be metabolized to HICA. So this question of whether this phenomenon is a consequence of just high accumulation of leucine, or whether this is truly a produced metabolite specific phenomenon becomes interesting and important. It can also help separate possible signaling roles of leucine vs metabolic roles, that cause this phenomenon.

8. The authors show that addition of glucose prevents the growth delay in cells shifted to CM (Figure 2A). Is this because of upregulating the same set of genes that help the cells cultured in HICA and 2K3MVA to reduce the growth delay (Figure 5A)? Can the authors determine if the major genes identified for adaptation in this study (SPCC1739.08c etc) are upregulated once glucose is added to the CM? This will strengthen the story and also address point number 1.

9. There is no data to establish unambiguously that the adaptation to autotoxins is epigenetic. Can it be just a transcriptional regulation where the adaptive genes are upregulated? Is this an inheritable phenotype (i.e. the resistant cells which grow remain resistant across generations)? This is the classical definition of epigenetic, and if this is not the case, then much of this can be just attributed to stochasticity in gene expression (and therefore heterogeneity). Is there a specific type of gene expression signature (that can come from stochastic events) that will provide resistance to cells?

10. The authors suggest that a few cells in the population adapt to the autotoxins epigenetically, possibly by upregulating genes for detoxification and toxin export. However, it is not clear whether these genes are specifically upregulated only in a few cells in the population. The data presented shows that the cells that are pre-exposed to toxins have these genes upregulated. However, it is possible that all the cells in CM have high expression of these genes, or there is stochasticity in gene expression (see point 7). Can the authors do conclusive experiments to determine if these genes are specifically upregulated in the few cells that have better survival in CM? One possibility is to use a fluorescent reporter under promoters of the major upregulated genes, and correlate the expression of the reporters with the cells that live/die. This would more unambiguously establish the phenomenon and conceptual mechanism that enables the phenomenon.

11. "Thus, the toxin producer is lost to a cheater, which only has the antitoxin system, whereas the cheater loses to cells having neither any toxins nor immunity [27]. In contrast, the voluntary triage does not cost much because the adaptation mechanism is usually offed without the toxin. This suggests that voluntary triage is resistant to cheaters" lines 187-191

In theory, cheaters which have upregulated genes for export and detoxification (the genes identified in this study) can survive the autotoxins without necessarily producing the autotoxins themselves. Can the authors clarify this? This goes back to the earlier question of whether the population actually splits into toxin producers vs non-producers/consumers, vs all cells producing (and sensing) the putative auto-toxins.

Minor points

1. The figures, especially the graphs can be labelled in more detail to help the readers understand the figures. The figures 3 B, C, D, E could be labelled in a better way. Since there are multiple plots in this paper, proper labelling will help the readers understand the figures and graphs better.

2. The title seems incomplete. There is some kind of autotoxin mediated triage….but? Triage leads to what? Triage exists? Is it really voluntary? I hope the authors give this some more thought.

Reviewer #2:

In this article, the authors describe a phenomenon where fission yeast release toxins (identified here using mass spectrometry as primarily leucic acid--2-hydrozyisocaproic acid, HICA and L-2keto-3methyvalerate--2K3MVA) under glucose starvation conditions. The discovery of HICA is interesting, as it has been shown to have antifungal and antimicrobial populations in other microbial studies, including of Lactobacillus. The authors term this toxin release voluntary triage. Cells that adapt to conditioned media (CM) containing the toxins can outcompete unadapted isogenic cells. The authors perform RNA-seq and find evidence that adaptation is due to expression of genes involved in transport (in addition the authors verified with genomic DNA-sequencing that survival in conditioned media was not due to accumulation of mutations). In addition, comparison with two S. cerevisiae strains indicates that these autotoxins and voluntary triage may be a conserved mechanism between distant yeast species. 

This is an interesting paper that may have implications for understanding microbial community dynamics, population dynamics, and multicellularity. However, I feel that the paper is underdeveloped and I do not have a good sense for the importance of the autotoxins in dictating population-level behavior (including heterogeneity in cell death) nor whether I should think about toxin production as a regulated process or a chance byproduct of metabolism under glucose-deprived conditions. I have some suggestions that should be addressed prior to publication:

Major Comments:

* The paper is written in a way that seems to imply the release of HICA and 2K3MVA are regulated processes or cellular "decisions", but is this an appropriate way to think about this process? I'm guessing that these molecules are being generated due to catabolism occurring in conjunction with starvation, so should I be thinking of this as anything more sophisticated than yeast having to take less desirable metabolic routes as the environment worsens, leading to the generation of toxic byproducts? 

* What is the mechanism of toxicity by the identified autotoxins? Why are the toxins only toxic in the absence of glucose? 

* Given that there seems to be the suggestion that toxin production and adaptation prevents mass-suicide (see Intro, Discussion) it would seem important to understand the nature of heterogeneity of entry into the stress-resistant states. The introductory sentences (line 17-22) would seem to suggest that as glucose is depleted cells differentiate into adapted and non-adapted cells, and the non-adapted cells will go on to die. Is there evidence for this? For example, how coherent is expression of the identified genes as cells progress into glucose starvation? 

* I find the choice of the term "voluntary triage" confusing. It seems to imply that cells are choosing to make toxic molecules. Or that this is an evolved mechanism. 

* How should I understand the similarity between Figure 1B and 1C? Since the fbp1 mutant cannot grow further in glycerol, when you switch to 0% MM are you adding more cells for fbp1 than for WT? Or are the fbp1 cells somehow producing the same amount of toxin over the many hours (up to 30 h) of incubation in 0% MM to produce CM despite presumably growing less in the 0% MM media? 

* Can you clarify how the search for the inhibitors was done? The methods say that the 12 were added at 10-40mM. Why was this the relevant concentration range? Particularly, given that later in the paper it looks like HIC and 2K3MVA were added at substantially lower concentrations.

* Its not clear to me that the statements made in line 180-183 are supported by the evidence. What counts as a mass suicide? Is there evidence that more cells would die without the production of the autotoxins? Similarly, the following comments about relative susceptibility to invasion by cheaters (lines 187-190) do not seem supported nor tested in the current work. 

Minor Comments:

* Line 15-It is not clear in this context what it means for cells to "tag" themselves. (Also line 67). 

* The shorthand for the media (3% MM is MM with both 3% glucose and 3% glycerol) is confusing. 

* Supplemental Figure 2: Does the delay phase really look the same? It seems longer without amino acids. 

* Figure 2: It is very hard to see some of the lines, especially the light blue line for fbp1CM+GLC (in both panel B and D)

* Line 112: Is the point of this line that the effective concentration of HIC and 2K3MVA required to inhibit cell growth is higher than one would expect given the mass spec data? The meaning of this sentence is not totally clear to me. And when the molecules are combined, how do the necessary inhibitory concentrations compare to the "natural condition"?

* Line 190: What does it mean "the adaptation mechanism is usually offed without the toxin?"

* Line 228-234—these methods are not clear

* Line 254—I think that the strain info is actually in Supplementary Table 5

* Line 266—missing Figure reference

* Why do so many of the growth curves show periodic fluctuations? (e.g. red line Supp Fig 1)

* this be? And secretion does not require growth? 

* Line 216—Reference 34 does not seem to be the correct reference here. 

Reviewer #3:

In this manuscript, the authors report an interesting phenomenon of autotoxin-mediated killing in yeasts. They show that switching the carbon source from glucose to glycerol results in the secretion of molecules that can kill yeast cells not previously exposed to the glycerol medium. This novel discovery opens an unexpected avenue into understanding intercellular communications in unicellular organisms.

I have the following suggestions for the authors to consider.

Major point

I am not fully convinced that HICA and 2K3MVA are the major components of autotoxins. The authors should provide measurements or estimation of the concentrations of HICA and 2K3MVA in the conditioned media as this information is important for interpreting the experiments performed using synthetic compounds. In addition, the authors did not rule out secreted proteins in the conditioned media as contributing factors. I suggest the authors test whether denaturing proteins by heating affects the cell-killing activity of conditioned media.

Minor points

(1) The term "voluntary triage"

I think the term "voluntary triage" gives an impression that a fraction of cells shifted to the glycerol medium are killed by autotoxins. But this is not be the case. Rather, cells that can be killed by autotoxins are naive "latecomer" cells not previously exposed to the glycerol medium. I suggest the authors consider using a different term, for example, "latecomer killing", to more accurately describe the phenomenon.

(2) The words "starved", "starvation", and "starving"

Title

"Autotoxin-mediated voluntary triage in starved yeast community"

Abstract

"we demonstrate an elaborate response of the yeast community against glucose starvation, named the voluntary triage"

"Glucose starvation" usually refers to shifting cells from glucose-containing medium to a medium lacking any carbon source. In this study, the condition used to induce autotoxins is shifting cells from glucose-containing medium to glycerol-containing medium. I think it is not appropriate to call this treatment "glucose starvation".

The authors may want to consider changing "starved yeast community" in the title to "yeast community adapting to a poor carbon source".

For the above sentence in the abstract, I suggest changing it to "we demonstrate an elaborate response of the yeast community to a shift to a poor carbon source and name this response the voluntary triage".

Elsewhere in the manuscript, the words "starved", "starvation", and "starving" also need to be re-considered.

(3) Are cells producing autotoxins sensitive to autotoxins?

Abstract

"yeast cells release autotoxins, such as leucic acid and L-2keto-3methylvalerate, which can even kill the cells producing them"

It is not clear to me how the authors can conclude that autotoxins kill the cells producing them. I think it is likely that cells producing autotoxins are resistant to killing.

(4) How are conditioned media prepared?

The following description is insufficient. 

"Cells were precultured in 3% MM and transferred to 0% MM and cultured. The supernatants were then filtered using a 0.22 μm PVDF membrane (Millex Sterile Filter Unit, Merck Millipore, Burlington, MA, USA)."

The authors should provide more details, including the starting OD600 in 0% MM and the OD600 at the 30 h time point. Because fbp1Δ cells do not grow in 0% MM, I wonder whether the authors might have used different starting OD600 for WT and fbp1Δ.

(5) The sources of synthetic autotoxin compounds

The authors should provide information on the sources from which enantiomers of HICA and 2K3MVA were obtained or how they were synthesized.

(6) Are genes important for adaptation to HICA also important for acquiring resistance in 0% MM?

It is useful to test whether genes such as SPAC57A7.05Δ are important for acquiring the ability to resist to conditioned media through preculturing in 0% MM.

(7) Are S. cerevisiae killed by auxotoxins?

I suggest the authors examine whether S. cerevisiae are killed by auxotoxins, for example, by performing a vital dye staining.

(8) A typo

Page 14

"cDNA libraries were prepared using the NEBNext Ultra II DNA Library Prep Kit 351 for Illumina (New England Biolabs, Inc. Ipswich, MA, USA), following the 352 manufacturer's instructions. cDNA libraries were sequenced using the Illumina 353 paired-end technology on MiSeq with MiSeq Reagent Kit v3"

In the above two sentences, I think "cDNA libraries" should be "genomic DNA libraries".

(9) How are HICA and 2K3MVA synthesized in the cells?

The authors should discuss how HICA and 2K3MVA may be synthesized in yeast cells.

Reviewer #4: 

The manuscript describes results of extensive studies examining the phenomenon of voluntary triage, an yeast response to glucose starvation. This response involves the production of autotoxins which serve double functions: while a subset of cells is killed by these autotoxins, the remaining cells adapt to their presence epigenetically and can grow efficiently under conditions of glucose starvation. 

The authors find convincing evidence for this phenomenon in the related yeast S. cerevisiae and argue for a conserved mechanism. This is a well done and solid study for which I only have a few suggestions to improve the quality of the manuscript.

1. One major aspect of the study that I see lacking is the understanding of why some cells become resistant to HICA and 2K3MVA while others do not. And whether the fate of these subpopulations is related in any way to their ability to produce these chemicals. So would cells that produce more HICA/2K3MVA are also more resistant to them. This could be tested by testing the MIC (minimum inhibitory concentration) of mutants defective in the production of HICA/2K3MVA.

2. How specific is this effect to glucose? It would be interesting to see if the absence of other fermentative sugars would lead to a similar effect (e.g. galactose, fructose, mannose etc).

3. Is the effect due to glucose signaling or glucose metabolism? This can be easily tested using 2-DOG. 

4. I found the discussion on multicellularity highly speculative and somewhat unrelated to the topic. Instead, I suggest the authors allocate more space to discussing population heterogeneity and the possible reasons for why some cells are killed by or adapt to the autotoxins and the evolutionary implications of this type of selection.

---

## [Decision Letter · Decision Letter 2]

9 Sep 2022

Dear Dr Hatakeyama,

Thank you for your patience while we considered your revised manuscript "Autotoxin-mediated latecomer killing in yeast community during glucose depletion" for publication as a Short Reports at PLOS Biology. This revised version of your manuscript has been evaluated by the PLOS Biology editors, the Academic Editor, and the original reviewers.

Based on the reviews, we are likely to accept this manuscript for publication, provided you satisfactorily address the remaining points raised by the reviewers. Please also make sure to address the following data and other policy-related requests.

IMPORTANT: Please attend to the following:

a) Please address the remaining requests from the reviewers.

b) Note that reviewers #1 and #2 both feel that some of your claims are "oversold" - please go through your manuscript carefully and ensure that your claims are commensurate with the supporting evidence.

c) Reviewer #1 also draws attention to some grammatical errors; we tend to agree, and recommend that you ask a native English-speaking colleague to read through your manuscript and correct these.

d) We recommend that you truncate your title to: "Autotoxin-mediated latecomer killing in yeast communities."

e) I can't find your supplementary Figures in the current version of the manuscript; please ensure that you include these!

f) Please address my Data Policy requests below; specifically, we need you to supply the numerical values underlying Figs 1BCEGHI, 2BCDEFGHI, 3GHIJKL, 4ABCDEFGH and all of the Supplementary Figures, either as a supplementary data file or as a permanent DOI’d deposition.

g) Please cite the location of the data clearly in all relevant main and supplementary Figure legends, e.g. “The data underlying this Figure can be found in S1 Data” or “The data underlying this Figure can be found in https://zenodoXXXXX”

We expect to receive your revised manuscript within two weeks. 

*Published Peer Review History*

*Press*

Sincerely,

Roland

Roland Roberts, PhD

Senior Editor,

rroberts@plos.org,

PLOS Biology

DATA POLICY:

Regardless of the method selected, please ensure that you provide the individual numerical values that underlie the summary data displayed in the following figure panels as they are essential for readers to assess your analysis and to reproduce it: Figs 1BCEGHI, 2BCDEFGHI, 3GHIJKL, 4ABCDEFGH and all of the Supplementary Figures. NOTE: the numerical data provided should include all replicates AND the way in which the plotted mean and errors were derived (it should not present only the mean/average values).

SPECIES INDICATED IN THE ABSTRACT? 

- Please note that per journal policy, the model system/species studied should be clearly stated in the abstract of your manuscript. 

DATA NOT SHOWN?

REVIEWERS' COMMENTS:

Reviewer #1:

[identifies himself as Sunil Laxman]

This version of a manuscript is a substantial improved and more readable version, primarily establishing a biochemical basis for killing cells within a clonal population of yeast. The use of 'latecomer-killing' also seems more appropriate than involuntary triage.

I enjoyed reading this version of the manuscript, and it is very effective as a short report. The manuscript needs a careful proofreading (to remove some minor grammatical errors, as well as slightly difficult to read sentences), but I will not suggest any further experiments. 

The last paragraph of the discussion (on the origins of multicellularity) is overly suggestive (especially with the liberal use of 'imply'). Of course, some speculation is encouraged in the discussion, but for example "The cell cycle dependency of cell death (Fig. S18) and the apoptotic pathway activation in a large fraction of dead cells (Fig. S17) also imply a relationship between latecomer killing and multicellularity." is quite optimistic, since it is still not clear how meaningful these data are with respect to say the artificial experiments forming snowflake yeasts. I would recommend a careful toning down of the last paragraph, but without eliminating the speculative element.

Reviewer #2: 

In this revision, the authors have converted their paper to a short report focused on the phenomenon of toxin release (identified here using mass spectrometry as primarily leucic acid--2-hydrozyisocaproic acid, HICA and L-2keto-3methyvalerate--2K3MVA) under glucose starvation conditions which they now term "latecomer killing". In addition, they have performed additional important and interesting experiments including measuring HICA and 2K3MVA concentration in conditioned media, assessing additional mutants, assessing additional nutrient add-backs, performing a 2-DOG experiment, analyzing cell death, and determining the effect of the cell cycle on killing. 

The manuscript is improved over the previous version and functions better as a short report. Indeed, the observation of the autotoxin phenomenon is extremely interesting and worth reporting. However, although the authors have performed additional experiments questions remain about whether the phenomenon provides a population-level advantage, why there is so much variability in the mechanism of cell death (if being caused by the same autotoxins), and mechanistically what is causing resistance or lack there-of. In addition, while the authors have dialed back some of their language the connections to cheater dynamics and multicellularity are tenuous at best. In addition, there is a tendency to oversell the results (for example, while the cell cycle data is intriguing it does not directly show that it is G0 cells that are resistant and indeed the idea that it is G0 cells also somewhat contradicts that idea that there is a dividing, resistant population.) Furthermore, lines 63-69 suggest that cells are inheriting an adapted state because their growth rate remains constant in CM. Since adapted cells are being inoculated into CM, I am unconvinced of this explanation. Of course, there is continuous pressure for cells to maintain the adapted state. That doesn't mean that they are necessarily remembering it as they divide, particularly in the epigenetic sense (although the authors did remove reference to epigenetics). 

This paper reports a very interesting phenomena that is deserving of publication with interesting experiments that hint at potential mechanisms and population-level effects. However, in some places the results are oversold and mechanistic insight is lacking. 

Reviewer #3:

The revision has addressed all my concerns. I support the publication of this paper.

Reviewer #4:

The authors have addressed most of the points raised by all 4 reviewers, either through additional experiments or through changes to the text.

Some issues could not be addressed due to the extensive work required which would be greatly beyond the scope of a Short Report. 

The results with the different sugars and the cell cycle mutant are interesting and worth pursuing.

---

## [Editor Report · Decision Letter 3]

22 Sep 2022

Dear Dr Hatakeyama,

Thank you for the submission of your revised Short Report "Autotoxin-mediated latecomer killing in yeast community" for publication in PLOS Biology. On behalf of my colleagues and the Academic Editor, Mark Siegal, I'm pleased to say that we can in principle accept your manuscript for publication, provided you address any remaining formatting and reporting issues. These will be detailed in an email you should receive within 2-3 business days from our colleagues in the journal operations team; no action is required from you until then. Please note that we will not be able to formally accept your manuscript and schedule it for publication until you have completed any requested changes.

Note: I have asked my colleagues to include the following request to you: "Thank you for changing the Title. However, please can you change the last word in the Title from 'community' to 'communities,' as previously requested?"

Sincerely, 

Roli Roberts

Senior Editor

PLOS Biology

rroberts@plos.org